# Hard-Attention for Scalable Image Classification

**Athanasios Papadopoulos**[1]    **Paweł Korus**[1,2]    **Nasir Memon**[1]
[1]Tandon School of Engineering, New York University
[2]AGH University of Science and Technology
{tpapadop, pkorus, memon}@nyu.edu

## Abstract

Can we leverage high-resolution information without the unsustainable quadratic complexity to input scale? We propose Traversal Network (TNet), a novel multi-scale hard-attention architecture, which traverses image scale-space in a top-down fashion, visiting only the most informative image regions along the way. TNet offers an adjustable trade-off between accuracy and complexity, by changing the number of attended image locations. We compare our model against hard-attention baselines on ImageNet, achieving higher accuracy with less resources (FLOPs, processing time and memory). We further test our model on fMoW dataset, where we process satellite images of size up to $896 \times 896$ px, getting up to $2.5$x faster processing compared to baselines operating on the same resolution, while achieving higher accuracy as well. TNet is modular, meaning that most classification models could be adopted as its backbone for feature extraction, making the reported performance gains orthogonal to benefits offered by existing optimized deep models. Finally, hard-attention guarantees a degree of interpretability to our model's predictions, without any extra cost beyond inference.

## 1    Introduction

In image classification, deep neural networks (DNNs) are typically designed and optimized for a specific input resolution, e.g. $224 \times 224$ px. Using modern DNNs on images of higher resolution (as happens e.g., in satellite or medical imaging) is a non-trivial problem due to the subtlety of scaling model architectures [57], and rapid increase in computational and memory requirements.

A linear increase in the spatial dimensions of the input, results in a quadratic increase in computational complexity and memory, and can easily lead to resource bottlenecks. This can be mitigated with careful engineering, e.g., streaming [46] or gradient checkpointing [42]. However, such solutions are content-agnostic, and don't take advantage of the fact that discriminative information may be sparse and distributed across various image scales, deeming processing of the whole input unnecessary.

Our goal is to leverage high-resolution information, while dispensing with the unsustainable quadratic complexity to input scale. To this end, we propose Traversal Network (TNet), a multi-scale hard-attention architecture, which traverses image scale-space in a top-down fashion, visiting only the most informative image regions along the way. TNet is recursive, and can be applied to inputs of virtually any resolution; an outline of its processing flow is presented in Fig. 1 (a). Our method draws its intuition from the way humans use saccades to explore the visual world.

TNet offers an adjustable trade-off between accuracy and complexity, by changing the number of attended image regions. This way, complexity increases linearly with the number of attended locations, irrespective of the input resolution. Also, hard-attention explicitly reveals the image regions that our model values the most, providing a certain degree of interpretability (Fig. 1 (c)). Importantly, interpretability comes without any extra cost beyond inference, in contrast to popular attribution methods, which require at least an additional backward pass [53], or numerous forward passes [1].

35th Conference on Neural Information Processing Systems (NeurIPS 2021).

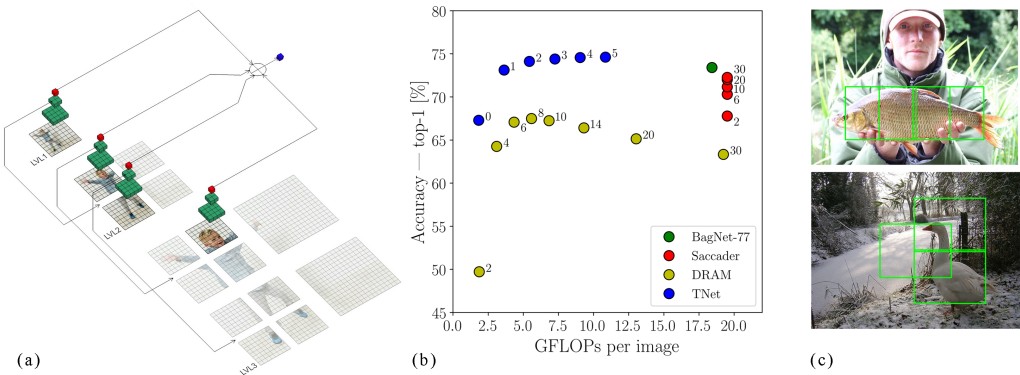

Figure 1: (a) Multi-scale processing in TNet. Starting at level 1, features are selectively extracted from image regions at various scales (red cubes), and then, they are combined to create the final image representation used for classification (blue cube). (b) Experimental results on ImageNet [13] with baselines based on [18]. Numeric annotations correspond to the number of attended locations. Our model offers a better trade-off between accuracy and complexity (FLOPs). (c) Examples of attention policy (top 3 locations) learned on ImageNet.

Attention may also reduce data acquisition cost [60], by allowing only a fraction of the high-resolution content to be acquired.

TNet is trained end-to-end by employing a modified version of REINFORCE rule [68], while using only classification labels. Our architecture is modular, and most classification models could be adopted as its backbone for feature extraction. This way, we can directly take advantage of various performance benefits offered by existing optimized deep models.

Hard-attention is the mechanism that allows TNet to dispense with quadratic complexity to input scale, and as a result, we evaluate our model against strong hard-attention baselines on ImageNet [18]. A summary of our results is depicted in Fig. 1 (b), where we see that TNet offers a better trade-off between accuracy and complexity measured in FLOPs (similar behavior is observed with actual timings and memory). We extend our experiments to fMoW dataset, which consists of high-resolution satellite images [12]. We process images up to $896 \times 896$ px, getting up to 2.5x faster processing compared to baselines operating on the same resolution, while achieving higher accuracy as well.

We find improvements in accuracy surprising, because TNet is processing only part of the input, in contrast to fully convolutional baselines. We primarily attribute this behavior to a novel regularization method, which encourages classification based on individual attended locations. We verify its efficacy through an ablation study.

## 2 Related work

**Attention.** Attention has a long history in the artificial neural networks literature [31], and in the modern era of deep learning it has been used very successfully in various problems [36, 14, 4, 70, 22, 67, 55]. Two main forms of attention are: *soft attention* which processes everything but weights various regions differently; and *hard attention* which selects only a fraction of the data for processing. Hard-attention models address various use-cases, and can be motivated by interpretability [18], reduction of high-resolution data acquisition cost [60], or computational efficiency [33]. Our goal is to offer a single model that takes advantage of all these benefits.

Our model is conceptually similar to glimpse-based models [2, 43, 50, 54, 19, 3]. An important difference is that we don't restrict our attention policy to 2D space, but we consider scale dimension as well. Also, our model parallelizes feature extraction at each processing level, instead of being fully sequential. Furthermore, we don't use recurrent neural networks (RNNs) to combine features from different locations, but instead, we simply average them (see Section 3). This way, gradients flow directly to extracted features, without the need to backpropagate through RNN steps. This is a simpler strategy compared to LSTM in order to avoid vanishing gradients.

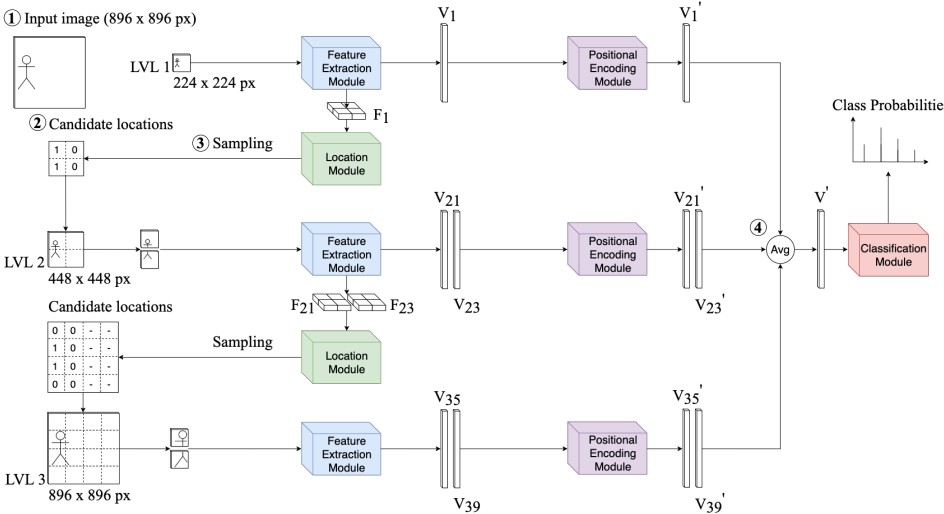

Figure 2: Three unrolled processing levels of our architecture. Starting at level 1, the image is processed in the coarsest scale (*Feature Extraction Module*), and the extracted features are used to decide which image locations should be processed in finer detail (*Location Module*). This process is repeated for each selected location to reach level 3, where features from the highest resolution are extracted. All features are enriched with positional information (*Positional Encoding Module*), and then are averaged before the final classification (*Classification Module*).

Recent work explores soft attention mechanisms based on transformers, which originate from the natural language processing community [63]. Transformers have already been used extensively in machine vision [45, 74, 16, 8, 48], and research interest in related directions has increased [6, 11, 5].

**Multi-scale representations.** We identify four broad categories of multi-scale processing methods. (1) *Image pyramid methods* extract multi-scale features by processing multi-scale inputs [17, 47, 34, 44]. Our model belongs to this category, and due to its recursive nature, it can extract features from an arbitrary number of pyramid levels (see Section 3). (2) *Encoding schemes* take advantage of the inherently hierarchical nature of deep neural nets, and reuse features from different layers, since they contain information of different scale [24, 41, 10]. (3) *Encoding-Decoding schemes* follow up the feed-forward processing (encoding) with a decoder, that gradually recovers the spatial resolution of early feature maps, by combining coarse and fine features [52, 40]. (4) *Spatial modules* are incorporated into the forward pass, to alter feature extraction between layers [71, 9, 65].

**Computational efficiency.** There are multiple ways to adjust the computational cost of deep neural networks. We organize them into four categories. (1) *Compression methods* aim to remove redundancy from already trained models [38, 26, 72]. (2) *Lightweight design strategies* are used to replace network components with computationally lighter counterparts [32, 30, 51, 27, 66]. (3) *Partial computation methods* selectively utilize parts of a network, creating paths of computation with different costs [37, 20, 73, 29, 69]. (4) *Attention methods* selectively process parts of the input, based on their importance for the task at hand [49, 39, 33, 56]. This is the strategy we follow in our architecture.

## 3 Architecture

### 3.1 Processing flow

We present our architecture by walking through the example in Figure 2, where we process an image with original resolution of $896 \times 896$ px (① in the top left corner). In the first level, we downscale the image to $224 \times 224$ px and pass it through the *feature extraction module*, in order to produce a feature vector $V_1$ that contains a coarse description of the original image.

To proceed to the next level, we feed an intermediate feature map, $F_1$, from the *feature extraction module* to the *location module*, which considers a number of candidate locations described by $F_1$,

and predicts their importance (in this particular example, the candidate locations form a $2 \times 2$ regular grid (②), and the *location module* yields 4 predictions). We express region importance as attendance probability, which parametrizes a categorical distribution used for sampling without replacement; in our current example, we sample 2 locations (③).

In the 2nd processing level, we crop the selected regions from the full-resolution image, resize them to $224 \times 224$ px, and feed them to the *feature extraction module* to obtain the corresponding feature vectors (here $V_{21}$ and $V_{23}$). The original input resolution allows us to move to a 3rd processing level, where we feed $F_{21}$ and $F_{23}$ to the *location module*, leading to 2 Categorical distributions. We sample 1 location from each one of them to get $V_{35}$ and $V_{39}$.

Features extracted from all levels are passed through the *positional encoding module*, which injects information about the spatial position and scale of the image regions the features describe. The resulting vectors, $\{V_*^{'}\}$, are averaged (④) into a single comprehensive representation, $V^{'}$, that is fed to the *classification module* for the final prediction.

## 3.2   Modules

The **feature extraction module** receives an image of fixed size as input, and outputs a feature vector $V$ and an intermediate spatial representation $F$. The input size, a hyperparameter we call *base resolution*, defines the minimum amount of information that can be processed. Hence, it constrains the minimum cost that our model has to pay in computational and memory resources. In our experiments, feature extraction modules are implemented using CNNs.

The **location module** predicts $K = n^2$ probabilities of a Categorical distribution over the locations within a given $n \times n$ grid of candidate image regions. It receives a feature map of size $n \times n \times c$ as input, where each $1 \times 1 \times c$ vector describes the corresponding location of the grid. The feature map is passed through a series of $1 \times 1$ convolutions (contextual information is infused as well, e.g., via squeeze and excitation [28]), yielding $K$ logits, which are transformed to relative region importance via a softmax layer.

The **positional encoding module** receives a feature vector $f$ and a positional encoding vector $p$, and combines them (e.g., through a fully connected layer) to an output feature vector $f^{'}$. We use a variant of the fixed positional encodings based on sine and cosine functions introduced in [63]. Instead of a single dimension of time, we have three: two spatial dimensions and scale.

The **classification module** projects the final feature vector (e.g., via a linear layer) to classification logits. We provide the exact module architectures in Appendix 2.1, along with justification of our design choices.

## 4   Training

### 4.1   Learning rule

Our model is not end-to-end differentiable because of location sampling. We address this problem using a variant of the *REINFORCE* [68] learning rule:

$$L_F = \frac{1}{N \cdot M} \sum_{i=1}^{N \cdot M} \left[ \frac{\partial \log p(y_i | l^i, x_i, w)}{\partial w} + \lambda_f (R_i - b) \frac{\partial \log p(l^i | x_i, w)}{\partial w} \right] \tag{1}$$

where $x_i$ is the $i$-th image, $y_i$ is its label, and $w$ are the parameters of our model. $p(l^i | x_i, w)$ is the probability that the sequence of locations $l^i$ is attended for image $x_i$, and $p(y_i | l^i, x_i, w)$ is the probability of predicting the correct label after attending to $l^i$.

$N \cdot M$ is the total number of examples used for each update. The size of our original batch $B$ is $N$, and we derive (1) using a Monte Carlo estimator with $M$ samples to approximate the expectation $\sum_{l^i} p(l^i | x_i, w) \left[ \frac{\partial \log p(y_i | l^i, x_i, w)}{\partial w} + \log p(y_i | l^i, x_i, w) \frac{\partial \log p(l^i | x_i, w)}{\partial w} \right]$ for each image $x_i$ in $B$. To reduce the variance of the estimator, we replace $\log p(y_i | l^i, x_i, w)$ with a discrete indicator function $R_i$, which is equal to 1 for correct predictions and 0 otherwise [2]. To the same end, we use baseline $b$, which corresponds to the exponential moving average of the mean reward $R_i$ $\forall i$, and is updated

after processing each training batch [70]:

$$b_n = 0.9 \cdot b_{n-1} + 0.1 \cdot \frac{1}{NM} \sum_{i=1}^{NM} R_i^n \qquad (2)$$

where $R_i^n$ is the reward for the $i$-th image in the $n$-th batch. For simplicity, we drop the subscript of $b_n$ in (1). $\lambda_f$ is a weighting hyperparameter. We provide a detailed derivation of our learning rule in Appendix 1.1.

The first term of $L_F$ is used to update the parameters in order to maximize the probability of the correct label. The second term is used to update the location selection process, according to the utility of the attended sequence $l^i$ in the prediction of the correct label.

## 4.2 Per-feature regularization

When our model attends to a location sequence $l_i$ and makes a correct prediction, we positively reinforce the probability to attend every location in the sequence. This is expressed in the second term of (1), where we use probability $p(l^i|x_i, w)$ of the whole sequence $l_i$. However, some locations may not contribute to the correct prediction, e.g., if they have missed the object of interest. In such cases, we reinforce the attendance of uninformative regions, encouraging a sub-optimal policy.

To mitigate this problem, for every attended location, we use its feature vector to make a separate classification prediction. Then, we use these predictions to complement our assessment on whether the corresponding image regions have useful content. We modify our learning rule as follows:

$$L_r = L_F^s + \frac{1}{|l^i|} \sum_{k=1}^{|l^i|} L_F^k, \qquad (3)$$

$$L_F^s = \frac{1}{N \cdot M} \sum_{i=1}^{N \cdot M} \left[ \lambda_c \frac{\partial \log p(y_i|l^i, x_i, w)}{\partial w} + \lambda_f \lambda_r (R_i^s - b) \frac{\partial \log p(l^i|x_i, w)}{\partial w} \right], \qquad (3a)$$

$$L_F^k = \frac{1}{N \cdot M} \sum_{i=1}^{N \cdot M} \left[ (1 - \lambda_c) \frac{\partial \log p(y_i|l_k^i, x_i, w)}{\partial w} + \lambda_f (1 - \lambda_r)(R_i^k - b) \frac{\partial \log p(l_k^i|x_i, w)}{\partial w} \right] \qquad (3b)$$

where $L_F^s$ is learning rule (1) with additional weighting hyperparameters $\lambda_c, \lambda_r \in [0, 1]$. $L_F^k$ is learning rule (1) when we attend only to the $k$-th location, $l_k^i$, from every sequence $l^i$. Also, we introduce weighting factors $(1 - \lambda_c)$ and $(1 - \lambda_r)$. During training, we attend to a fixed number of $|l^i|$ locations for every image. $R_i^s$ and $R_i^k$ are discrete indicator functions equal to 1 when a prediction is correct, and 0 otherwise. $L_F^s$ updates the parameters of our model based on attending to $l^i \ \forall i$, while $L_F^k$ updates them based on $l_k^i$; $\lambda_c$ and $\lambda_r$ specify the relative importance of these updates.

Even though our initial motivation was to improve the attention policy, the first term in (3b) updates feature extraction parameters based on independent predictions from attended image regions. We empirically observed that such updates boost performance, potentially because they lead to features that co-adapt less and generalize better.

## 5 Experimental evaluation

### 5.1 Effectiveness of hard-attention mechanism

**Data.** ImageNet [13] consists of natural images from $1,000$ classes. We use the ILSVRC 2012 version, which consists of $1,281,167$ training and $50,000$ validation images.

**Models.** We use Saccader and DRAM [18] as hard-attention baselines. We use Saccader with BagNet-77-lowD [7] as its backbone for feature extraction. BagNet-77-lowD is based on ResNet-50 [25], with receptive field constrained to $77 \times 77$ px. DRAM uses the standard ResNet-50 for feature extraction, with glimpses of $77 \times 77$ px.

For fair comparison, we set the base resolution of TNet to $77 \times 77$ px, and use BagNet-77, a slightly modified version of BagNet-77-lowD, as the feature extraction module. In the location module, we

Table 1: Efficacy of hard-attention mechanism: TNet surpasses all hard-attention baselines [18] on ImageNet, by attending to just 1 location. Higher accuracy can be achieved for less FLOPs, and translates to lower actual run time. Memory savings are obtained compared to BagNet-77 and Saccader as well. TNet has slightly more parameters compared to BagNet-77 due to the additional modules, but has significantly fewer parameters than the hard-attention baselines. FLOPs, run time, and memory are measured during inference.

| Model | #Locs | Top-1 Acc. | Top-5 Acc. | FLOPs (B) | #Params (M) | Time (msec/im) | Memory (GB) |
|---|---|---|---|---|---|---|---|
| **Saccader** | 2 | 67.79% | 85.42% | 19.5 | | 7.31 ± 1.55 | 2.58 |
| | 6 | 70.31% | 87.8% | 19.5 | 35.58 | 7.32 ± 1.55 | 2.52 |
| | 20 | 72.02% | 89.51% | 19.51 | | 7.36 ± 1.50 | 2.53 |
| | 30 | 72.27% | 89.79% | 19.51 | | 7.36 ± 1.49 | 2.51 |
| **DRAM** | 2 | 49.72% | 73.27% | 1.86 | | 3.43 ± 1.57 | 0.45 |
| | 4 | 64.26% | 84.84% | 3.1 | 45.61 | 3.92 ± 1.56 | 0.44 |
| | 8 | 67.5% | 86.6% | 5.58 | | 4.61 ± 1.59 | 0.45 |
| | 20 | 65.15% | 84.58% | 13.03 | | 7.58 ± 1.53 | 0.46 |
| **BagNet-77** | - | 73.42% | 91.1% | 18.42 | 20.55 | 5.94 ± 0.09 | 2.62 |
| **TNet** | 0 | 67.29% | 87.38% | 1.82 | | 0.74 ± 0.01 | 0.46 |
| | 1 | 73.12% | 90.56% | 3.63 | | 1.43 ± 0.01 | 0.57 |
| | 2 | 74.12% | 91.18% | 5.43 | 21.86 | 2.09 ± 0.02 | 0.69 |
| | 3 | 74.41% | 91.4% | 7.24 | | 2.74 ± 0.03 | 0.95 |
| | 5 | 74.62% | 91.35% | 10.84 | | 3.96 ± 0.04 | 1.47 |

use a uniform $5 \times 5$ grid of overlapping candidate regions. The dimensions of each grid cell span $34.375\%$ of the corresponding image dimensions. This way, for an image of size $224 \times 224$ px, the image patches within the grid cells at the 2nd processing level are $77 \times 77$ px. We use BagNet-77 as a separate fully convolutional baseline, with inputs of $224 \times 224$ px. Additional details about TNet and BagNet-77 are provided in Appendix 2.1.1.

**Training.** We train TNet with 2 processing levels on images of $224 \times 224$ px using class labels only. We train for 200 epochs using the Adam optimizer [35] with initial learning rate $10^{-4}$, that we drop once by a factor of $0.1$. We use dropout (keep probability $0.5$) in the last layer of feature extraction. We use per-feature regularization with $\lambda_c = \lambda_r = 0.3$. We attend to a fixed number of 3 locations.

We train BagNet-77 from scratch, on $224 \times 224$ px images. Compared to TNet, we reduce dropout keep probability to $0.375$, and we early-stop at $175$ epochs. We don't train our own Saccader and DRAM, we use results reported in [18]. Additional training details are provided in Appendix 2.2.1.

**Results.** We present our results in Table 1. TNet outperforms Saccader and DRAM by attending to only 1 location, while it surpasses BagNet-77 by attending to 2. Saccader was designed with accuracy and interpretability in mind, which leads to sub-optimal computational efficiency, as it processes the entire full-resolution image before attending to locations. As a result, FLOPs stay nearly constant and remain similar to BagNet-77, more that $5$ times higher than TNet with 1 attended location. DRAM offers the expected gradual increase in computational cost as the number of attended locations increases, but for FLOPs comparable with TNet - e.g., DRAM for 8 locations (maximum accuracy) and TNet for 2 - our model is superior by more than $6.5\%$.

TNet has slightly more parameters than BagNet-77, because of the location and positional encoding modules. Saccader and DRAM have significantly heavier attention mechanisms in terms of parameters.

We profile all models to validate correspondence between theoretical FLOPs and real processing speed, and to assess memory requirements. We time inference on batches of $64$ images; early batches are discarded to discount code optimization. We use a single NVIDIA Quadro RTX 8000 GPU, with $64$ GB of RAM, and 20 CPUs to mitigate data pipeline impact. For Saccader and DRAM we use public implementations [21] in TensorFlow (TF) 1. TNet and BagNet-77 are implemented in TF 2. The difference in TF versions may be a confounding factor in the obtained results. However, TNet and BagNet-77 use the same environment and yield the expected difference in latency. DRAM attends to locations sequentially, while TNet processes all locations from the same level in parallel.

Table 2: TNet effectively scales to images of resolution $448 \times 448$ px and $896 \times 896$ px on fMoW [12]. It surpasses in accuracy EfficientNet-B0 baselines trained on inputs of the same resolution, while it requires less FLOPs. Differences in FLOPs translate to differences in actual run time, while memory requirements are lower as well. TNet has more parameters compared to EficientNet-B0 due to the additional modules. FLOPs, run time, and memory are measured during inference. A graphical representation of the main results is provided in Appendix 2.4.1.

| Model | Input Size | #Locs | BBoxes | Top-1 Acc. | Top-5 Acc. | FLOPs (B) | #Params (M) | Time (msec/im) | Memory (GB) |
|---|---|---|---|---|---|---|---|---|---|
| **EfficientNet-B0** | $224^2$ | - | $\checkmark$ | 69.7% | 89.22% | 0.39 | 4.13 | $0.80 \pm 0.01$ | 0.76 |
| **ResNet-50** [60] | $224^2$ | - | $\checkmark$ | 67.3% | - | 4.09 | 23.71 | - | - |
| **DenseNet-121** [61] | $224^2$ | - | $\checkmark$ | 70.7% | - | 3 | 7.1 | - | - |
| | $224^2$ | | | 62.8% | 84.97% | 0.39 | | $0.80 \pm 0.01$ | 0.76 |
| **EfficientNet-B0** | $448^2$ | - | - | 69.83% | 90.22% | 1.54 | 4.13 | $3.04 \pm 0.18$ | 2.6 |
| | $896^2$ | | | 70.6% | 90.81% | 6.18 | | $12.18 \pm 0.18$ | 10.35 |
| | $224^2$ | 0 | - | 47.79% | 81.14% | 0.39 | | $0.84 \pm 0.02$ | 0.76 |
| **TNet** | | 1 | | 70.17% | 90.99% | 0.77 | | $1.74 \pm 0.02$ | 1.3 |
| | $448^2$ | 2 | - | 71.46% | 91.58% | 1.16 | 4.56 | $2.46 \pm 0.03$ | 1.55 |
| | | 3 | | 71.57% | 91.72% | 1.55 | | $3.18 \pm 0.04$ | 2.03 |
| | $896^2$ | 4 | - | 72.16% | 91.98% | 1.94 | | $4.71 \pm 0.03$ | 3.13 |
| | | 6 | | 71.92% | 91.83% | 2.71 | | $6.13 \pm 0.05$ | 3.47 |

This leads to fixed memory use in DRAM and monotonic increase in TNet, as the number of attended locations increases. We could trade this for latency by switching to fully sequential processing.

## 5.2 Scalability

**Data.** Functional Map of the World (fMoW) [12] consists of high-resolution satellite images from 62 classes. They are split in $363,572$ training, $53,041$ validation and $53,473$ testing images. All images have bounding box annotations.

**Models.** We use EfficientNet-B0 (EN-B0) [57] as the feature extraction module of TNet, with base resolution of $224 \times 224$ px. In the location module we use a $3 \times 3$ grid with overlapping cells ($50\%$ overlap). Hence, when processing 2 levels, we use inputs of $448 \times 448$ px, and with 3 levels, we use inputs of $896 \times 896$ px. Details of the TNet architecture are provided in Appendix 2.1.2.

We create our first 3 baselines by training EN-B0 with images resized to different resolution; $224 \times 224$ px, $448 \times 448$ px, and $896 \times 896$ px. We use the available bounding boxes to crop the regions of interest, resize them to $224 \times 224$ px, and train an additional EN-B0 baseline. Also, we report the best accuracy we find in the literature, achieved by single-model predictions (not network ensembles), and without the use of any meta-data for training. These are the ResNet-50 [60] and DenseNet-121 [61] baselines.

**Training.** We train TNet using only classification labels. The training proceeds in 2 stages: (1) training from scratch for 40 epochs on $448 \times 448$ px inputs, using 2 processing levels, and a fixed number of 2 locations; (2) fine-tuning for 10 more epochs on images of $896 \times 896$ px, which allow us to extend processing to 3 levels. We use 4 locations; 2 in the 2nd processing level, plus 1 for each of them in the 3rd. We use per-feature regularization in both stages. All 4 EN-B0 baselines are trained independently, from scratch. The exact hyperparameters are provided in Appendix 2.2.2.

**Results.** We present our results in Table 2. TNet surpasses all baselines by attending to 2 locations (2 processing levels), with top-1 accuracy of $71.46\%$. This requires less FLOPs than the corresponding EN-B0 baseline at the same input size ($448 \times 448$ px), which achieves accuracy of $69.83\%$.

Extending processing to 3 levels with 4 attended locations, further increases accuracy to $72.16\%$, while FLOPs increase by $\sim 67\%$ (from 1.16 to 1.94B FLOPs). The corresponding EN-B0 baseline operating on $896 \times 896$ px inputs, achieves $70.6\%$ accuracy, and requires $\sim 300\%$ more FLOPs compared to EN-B0 operating on $448 \times 448$ px inputs (from 1.54 to 6.18B FLOPs). This shows that TNet can efficiently leverage high-resolution information.

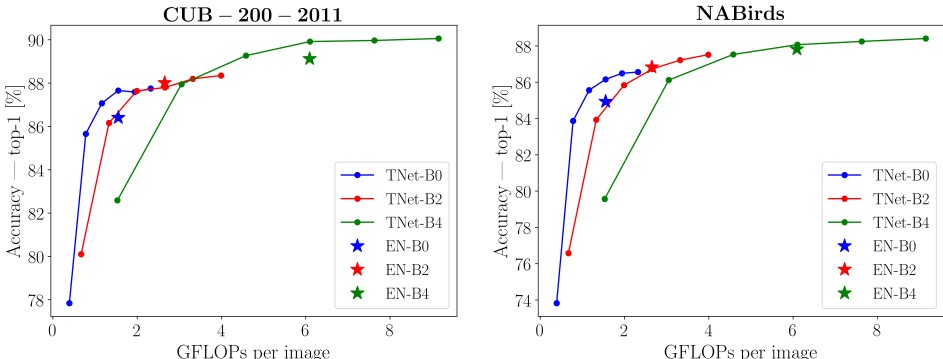

Figure 3: TNet is modular, and different models can be used as its feature extraction module. We plot TNet performance with 3 different backbones (attended locations range from 0 to 5). Stronger backbones lead to the expected increase in accuracy and FLOPs. EN baselines and the feature extraction modules of TNet models are initialized with pre-trained weights, while the rest of the modules are randomly initialized. TNet models achieve similar or better accuracy compared to corresponding baselines, showing that fine-tuning can be an effective practice.

We profile our models by following the procedure described in Section 5.1, and we show that differences in FLOPs translate to differences in actual run time. TNet also requires less memory, although it has more parameters because of the location and positional encoding modules.

We note that when TNet limits its processing to just 1 level (it uses only the contextual feature vector, $f_c$, extracted from the downscaled version of the whole image), its accuracy is considerably lower compared to EN-B0 operating on $224 \times 224$ px inputs. We hypothesize that as TNet is trained with more processing levels, $f_c$ co-adapts with the increasing number of extracted feature vectors. To mitigate this and maintain high accuracy at every processing level, as part of our future work, we can update Equation 3b by appropriately weighting terms that correspond to feature vectors from different levels.

### 5.3 Modularity and fine-tuning

TNet is a modular architecture, and we would like to test its compatibility with different backbone models, especially when these models are initialized with pre-trained weights, since this is a popular practice [15, 16, 58].

**Data.** We use 2 fine-grained classification datasets, CUB-200-2011 [64] and NAbirds [62], which are commonly used as downstream tasks. CUB-200-2011 has 200 classes of bird species, with $5,994$ training and $5,794$ testing images. NABirds has 555 classes of bird species, with $23,929$ training and $24,633$ testing images. We resize all images to $448 \times 448$ px.

**Models.** We use baselines from the EfficientNet (EN) family [57], EN-B$i$, $i \in \{0, 1, ..., 4\}$. We use the same EN models as the feature extraction module of TNet, getting TNet-B$i$. All TNet models have base resolution of $224 \times 224$ px, attention grid of $5 \times 5$, and each grid cell dimension occupies $35\%$ of the corresponding image dimension. Processing extends to 2 levels.

For TNet models, we use a weighted average to replace the simple averaging of the extracted feature vectors before the final prediction (④ in Fig. 2). The weights for the averaging are calculated by a new *feature weighting module*, which receives the $N$ feature vectors extracted from all processing levels, and outputs $N$ weights that sum up to $1$. Details about all architectures are provided in Appendix 2.1.3.

**Training.** EN baselines and the feature extraction modules of TNet models are initialized with weights pre-trained on ImageNet [59]. The rest of TNet modules are randomly initialized. All models are fine-tuned on the downstream tasks. Training details are provided in Appendix 2.2.3.

**Results.** We summarize our results in Figure 3. For clarity, we limit the number of models we plot, and we provide detailed results in Appendix 2.4.2. We see that performance differences between baselines, translate to similar differences between TNet models with corresponding backbones,

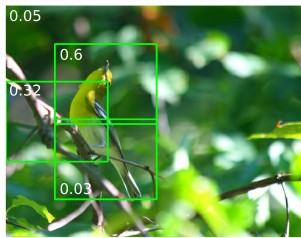 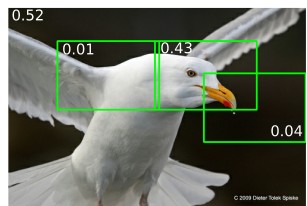 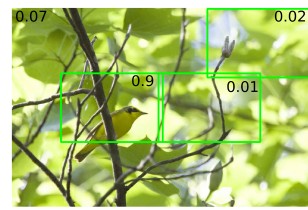

Figure 4: Examples of attention policy learned on CUB-200-2011. Numeric annotations correspond to weights predicted by the *feature weighting module* for the top 3 locations and the downscaled version of the whole image (1st processing level). Weights sum up to 1.

e.g., EN-B4 and TNet-B4 achieve considerably better accuracy compared to EN-B0 and TNet-B0 respectively. This indicates that behavioral differences between classification networks can be manifested in TNet models as well, if these networks are used as feature extraction modules.

In addition, as attended locations increase, TNet models achieve similar or better accuracy compared to corresponding baselines (TNet-B4 is also competitive to strong baseline API-Net [75] and state-of-the-art TransFG [23]; see Appendix 2.4.2). This indicates that initialization with pre-trained weights allows TNet to learn useful weights for all modules.

## 5.4 Attention policy and interpretability

We show examples of attended locations in Fig. 1 (c) and Fig. 4; more examples from all datasets are provided in Appendix 3.2. On ImageNet, the learned policy is highly content dependent and informative, as it overlaps with intuitively meaningful regions. The same behavior is observed in CUB-200-2011 and NABirds datasets, where the weights estimated by the *feature weighting module* have significant contribution to the interpretability of the predictions. On fMoW, our model predominantly attends to locations at the center, where nearly all objects of interest are located. While this is a correct heuristic implied by the data, it leads to a biased attention policy that is not sensitive to content changes. This demonstrates that sufficient diversity is needed to train fully informative policies.

To examine the relative importance between the coarse context and the information from the attended locations, we evaluate TNet on ImageNet, by using features only from attended locations. With 2, 3 and 5 attended locations, TNet achieves top-1 accuracy 67.95%, 69.66% and 71.05% respectively (with contextual information included, the corresponding accuracy values are 74.12%, 74.41% and 74.62%). The Saccader achieves similar accuracy for 2, 4 and 8 locations, with 67.8%, 69.51% and 70.08%. This shows that features extracted from attended locations have discriminative value even without global context. As a result, the attention policy can be valuable for interpreting predictions.

Hard-attention can also be used for reduction of high-resolution data acquisition cost [60]. On fMoW, when our model attends to 4 locations in 3 processing levels, it covers 37.91% of the input in resolution of $448 \times 448$ px, and less than 12.5% in the highest resolution of $896 \times 896$ px. Further quantitative analysis of the attention policy is provided in Appendix 3.1.

## 5.5 Ablation study

We examine the effect of per-feature regularization by training TNet on ImageNet without it. With 1 attended location, top-1 accuracy drops from 73.12% (Table 1) to 65.21%, while with 3 and 5 locations, accuracy drops from 74.41% and 74.62%, to 67.33% and 68.55% respectively. The drop is substantial, placing TNet below both Saccader and BagNet-77 in terms of accuracy.

Per-feature regularization may have similar impact as cropping-based data augmentation, since it forces the model to make independent predictions with features from every attended location. However, the attention policy is not random, but learned, which is crucial for the quality of the crops. In addition, we don't get one image crop per epoch, but multiple crops in the same training iteration. We hypothesize that this is important to prevent feature co-adaptation, since the model learns to recognize the same object from multiple crops simultaneously.

# 6 Conclusion

We proposed a novel multi-scale hard-attention architecture, TNet, that can efficiently scale to images of high resolution. By controlling the number of attended locations, TNet can adjust the accuracy-computation trade-off dynamically. We demonstrated the efficacy of our method on ImageNet against strong hard-attention baselines, and we further verified its behavior with high-resolution satellite images (fMoW). The attention policy reveals the image regions deemed more informative by our model, and makes its predictions inherently interpretable.

There are multiple research directions that can address current limitations of our method. First, we would like the decision on the number of attended locations to stem from a content-dependent learned policy. In addition, we would like scale-space traversal to be bi-directional, instead of merely top-down, in order for processing to be more adaptive. To the same end, we would like already extracted features to condition the processing of subsequent locations.

On the broader impact of our approach, we hypothesize that under a number of assumptions, hard-attention has the potential to be useful to any intelligent agent that navigates through the immense complexity of the natural visual world. These assumptions are that (1) the available resources are limited (2) there are performance constraints e.g. maximum response time, (3) the available information is practically infinite (4) all bits of information are not equally useful, and can even be misleading, e.g. noise. In this context, it would be beneficial for an intelligent agent to prioritize the expenditure of its resources according to the utility of the available information, in order to reach its performance goals; this is what a learned hard-attention mechanism can facilitate.

## Funding Transparency Statement

The authors did not receive third-party funding or third-party support for their research work.

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
