# Appendix - Hard-Attention for Scalable Image Classification

**Athanasios Papadopoulos**[1]    **Paweł Korus**[1,2]    **Nasir Memon**[1]

[1]Tandon School of Engineering, New York University
[2]AGH University of Science and Technology
{tpapadop, pkorus, memon}@nyu.edu

## 1   Training

### 1.1   Learning rule derivation

The *REINFORCE* rule naturally emerges if we optimize the log likelihood of the labels, while considering the attended locations as latent variables [1]. Given a batch of $N$ images, for the log likelihood we get:

$$\sum_{i=1}^{N} \log p(y_i|x_i, w) = \sum_{i=1}^{N} \log \sum_{l^i} p(l^i|x_i, w) p(y_i|l^i, x_i, w) \tag{1}$$

where $x_i$ is the $i$-th image in the batch, $y_i$ is its label, and $w$ are the parameters of our model. $p(l^i|x_i, w)$ is the probability that the sequence of locations $l^i$ is attended for image $x_i$, and $p(y_i|l^i, x_i, w)$ is the probability of predicting the correct label after attending to $l^i$. Equation 1 describes the log likelihood of the labels in terms of all location sequences that could be attended. $p(y_i|l^i, x_i, w)$ is computed by the classification module, and $p(l^i|x_i, w)$ is computed by the location module (see Section 1.2).

We use Jensen's inequality in Equation 1 to derive the following lower bound on the log likelihood:

$$\sum_{i=1}^{N} \log p(y_i|x_i, w) \geq \sum_{i=1}^{N} \sum_{l^i} p(l^i|x_i, w) \log p(y_i|l^i, x_i, w) = F \tag{2}$$

By maximizing the lower bound $F$, we expect to maximize the log likelihood. The update rule we use, is the partial derivative of $F$ with respect to $w$, normalized by the number of images in the batch. We get:

$$\frac{1}{N}\frac{\partial F}{\partial w} = \frac{1}{N} \sum_{i=1}^{N} \sum_{l^i} \left[ p(l^i|x_i, w)\frac{\partial \log p(y_i|l^i, x_i, w)}{\partial w} + \log p(y_i|l^i, x_i, w)\frac{\partial p(l^i|x_i, w)}{\partial w} \right] \Rightarrow$$

$$\frac{1}{N}\frac{\partial F}{\partial w} = \frac{1}{N} \sum_{i=1}^{N} \sum_{l^i} p(l^i|x_i, w)\left[ \frac{\partial \log p(y_i|l^i, x_i, w)}{\partial w} + \log p(y_i|l^i, x_i, w)\frac{\partial \log p(l^i|x_i, w)}{\partial w} \right] \tag{3}$$

To derive (3), we used the log derivative trick. As we can see, for each image $x_i$ we need to calculate an expectation according to $p(l^i|x_i, w)$. We approximate each expectation with a Monte Carlo estimator of $M$ samples:

$$\frac{1}{N}\frac{\partial F}{\partial w} \approx \frac{1}{N}\frac{\partial \tilde{F}}{\partial w} = \frac{1}{N} \sum_{i=1}^{N} \frac{1}{M} \sum_{m=1}^{M} \Big[ \frac{\partial \log p(y_i|l^{i,m}, x_i, w)}{\partial w} +$$

$$\log p(y_i|l^{i,m}, x_i, w)\frac{\partial \log p(l^{i,m}|x_i, w)}{\partial w} \Big] \tag{4}$$

35th Conference on Neural Information Processing Systems (NeurIPS 2021).

$l^{i,m}$ is the sequence of locations attended during the $m$-th sample from $p(l^i|x_i, w)$ (we get samples by repeating the processing of image $x_i$).

In order to reduce the variance of the estimator, we replace $\log p(y_i|l^{i,m}, x_i, w)$ with a reward function $R_{i,m}$, which is equal to 1 when the prediction for $x_i$ in the $m$-th sample is correct, and 0 otherwise. In addition, we use the baseline technique from [26], which corresponds to the exponential moving average of the mean reward $R_{i,m}$ $\forall i, m$, and is updated after processing each training batch. Our baseline is initialized to 0.5, and after the $n$-th batch we get:

$$b_n = 0.9 \cdot b_{n-1} + 0.1 \cdot \frac{1}{NM} \sum_{i=1}^{NM} R_i^n \tag{5}$$

where $R_i^n$ is the reward for the $i$-th image in the $n$-th batch. Since we use $M$ samples for the Monte Carlo estimator of each image, we simply consider that our batch has size $NM$ to simplify notation. Our learning rule (4) is updated as follows:

$$L_F = \frac{1}{NM} \sum_{i=1}^{NM} \left[ \frac{\partial \log p(y_i|l^i, x_i, w)}{\partial w} + \lambda_f (R_i - b) \frac{\partial \log p(l^i|x_i, w)}{\partial w} \right] \tag{6}$$

For simplicity, we drop the subscript of $b_n$ that indicates the batch we are processing. Also, we add a weighting hyperparameter $\lambda_f$. Equation 6 is the learning rule we presented in Section 4.1. of our paper, and this concludes our derivation.

## 1.2 Sampling approximation

In order to attend to a sequence of locations $l^i$, we sample without replacement from a series of Categorical distributions. For the probability of attending to a sequence $l^i$, we get:

$$p(l^i|x_i, w) = \prod_{j=1}^{N^{l^i}} \prod_{r=1}^{L_j^{l^i}} \prod_{k=1}^{g} \left[ \frac{p_j^{l^i}(l_k|x_i, w)^{u_{j,k,r}^{l^i}}}{\sum_{k=1}^{g} \left[ p_j^{l^i}(l_k|x_i, w) \prod_{r'=1}^{r-1} \left(1 - u_{j,k,r'}^{l^i}\right) \right]} \right] \tag{7}$$

where $N^{l^i}$ is the number of Categorical distributions (equal to the number of times the location module is applied), $L_j^{l^i}$ is the number of samples we draw from the $j$-th distribution, and $g$ is the total number of candidate locations per distribution. In the example of Fig. 2 in our paper, we consider 3 distributions ($N^{l^i} = 3$), $L_1^{l^i} = 2$ in the 2nd processing level and $L_2^{l^i} = L_3^{l^i} = 1$ in the 3rd, and $g = 4$ since we consider a $2 \times 2$ grid.

$l_k$ is the $k$-th out of the $g$ candidate locations, and $p_j^{l^i}(l_k|x_i, w)$ is the probability of selecting $l_k$ in the $j$-th distribution. $u_{j,k,r}^{l^i}$ is an indicator function that is equal to 1 when location $l_k$ is attended as the $r$-th sample of the $j$-th distribution, and 0 otherwise. $p_j^{l^i}(l_k|x_i, w)$ is computed by the location module, and $u_{j,k,r}^{l^i}$ is the outcome of sampling from the $j$-th Categorical distribution. The denominator in (7) is applicable for $r > 1$, and normalizes the probabilities of the $j$-th Categorical distribution before the $r$-th sample, to account for the lack of replacement.

In order to simplify our implementation of sampling dictated by (7), we introduce two modifications. First, we approximate sampling by selecting the locations with the $L_j^{l^i}$ highest probabilities. Potential downside is that we miss the opportunity to attend to less probable locations that may have valuable information (less exploration). However, at the beginning of training, all locations start with practically equal probability, and even by picking the top $L_j^{l^i}$ locations, we are able to explore the location space.

Second, we disregard the normalization factor for each $p_j^{l^i}(l_k|x_i, w)^{u_{j,k,r}^{l^i}}$ (denominator in (7)). This simplification does not affect the relative ordering between the probabilities of each Categorical distribution. As a result, the $L_j^{l^i}$ locations with the highest probabilities that we attend to, remain the same.

Table 1: Building blocks of our architectures. ConvBlock and MBConv$F$ have residual connections that add the input to the output. If $s > 1$, or the number of input channels is not equal to the output channels, MBConv$F$ drops the residual connection. If the same conditions hold true for the ConvBlock, it applies an $1 \times 1$ convolution with stride $s$ and channels $C$ to the input before it is added to the output. Also, if $p$ is VALID, a total margin of $k - 1$ pixels is dropped from each spatial dimension of the input, before it is passed through the residual connection. The first layer in MBConv$F$ is performed only if $F \neq 1$. Both for SE-$(C_r, \ r)$ and MBConv$F$, $C_{in}$ corresponds to the number of input channels and is not a parameter of the blocks. Batch Norm [13] is applied before the activation. GAP stands for Global Average Pooling, and DWConv for depthwise convolution.

| Block Type | Layer/Block Type | Kernel Size | #Output Channels | Stride | Padding | Batch Norm | Activation |
|---|---|---|---|---|---|---|---|
| **ConvBlock** [8] | Conv | $1 \times 1$ | $C/4$ | 1 | SAME | - | Leaky ReLU |
| | Conv | $k \times k$ | $C/4$ | $s$ | $p$ | - | Leaky ReLU |
| | Conv | $1 \times 1$ | $C$ | 1 | SAME | - | Leaky ReLU |
| **Squeeze and Excitation-$(C_r, \ r)$ SE-$(C_r, \ r)$** [10] | GAP | - | $C_{in}$ | - | - | - | - |
| | Conv | $1 \times 1$ | $C_r \cdot r$ | 1 | SAME | - | SiLU |
| | Conv | $1 \times 1$ | $C_{in}$ | 1 | SAME | - | Sigmoid |
| | Multiply | - | $C_{in}$ | - | - | - | - |
| **MBConv$F$** [18] | Conv (if $F \neq 1$) | $1 \times 1$ | $C_{in} \cdot F$ | 1 | SAME | $\checkmark$ | SiLU |
| | DWConv | $k \times k$ | $C_{in} \cdot F$ | $s$ | SAME | $\checkmark$ | SiLU |
| | SE-$(C_{in}, \ 0.25)$ | - | $C_{in} \cdot F$ | - | - | - | - |
| | Conv | $1 \times 1$ | $C$ | 1 | SAME | $\checkmark$ | - |

## 2 Experimental evaluation

### 2.1 Architectures

We first present the architectures we use in our experiments (Section 5 in our paper), and then we provide more details about the design of individual modules. In Table 1, we provide the building blocks of our architectures.

#### 2.1.1 Models used on ImageNet

In Table 2, we provide the TNet architecture we use in our experiments on ImageNet [4] (Section 5.1 in our paper). BagNet-77 baseline corresponds to TNet's feature extraction module followed by the classification module.

BagNet-77 results from BagNet-77-lowD with 3 modifications. First, we replace "VALID" padding of some convolutional layers with "SAME", to obtain less aggressive reduction of the spatial dimensions; the base resolution of TNet is $77 \times 77$ px, instead of $224 \times 224$ px which is the input size of the Saccader's backbone. Second, we remove Batch Normalization due to technical issues in preliminary experiments (Batch Norm was successfully used in later experiments with the other datasets). Third, we use Leaky ReLU instead of ReLU activations, to allow non-zero gradients for negative inputs.

#### 2.1.2 Models used on fMoW

In Table 3, we provide the TNet architecture we use in our experiments on fMoW [2] (Section 5.2 in our paper). EfficientNet-B0 baseline corresponds to TNet's feature extraction module followed by the classification module.

#### 2.1.3 Models used on CUB-200-2011 and NABirds

We get the feature extraction module of each TNet-B$i$, $i \in \{0, 1, ..., 4\}$, by removing the last fully connected layer of the corresponding EN-B$i$ model; this happens in Table 3 as well, where we get the feature extraction module of TNet from EN-B0. The location and positional encoding modules are implemented as in Table 3, with output channels scaled according to the feature extraction module in use. For the location module, the attention grid is $5 \times 5$, leading to an output of size $1 \times 25$. Also, for different TNet-B$i$ models, the input to the location module may vary in number of channels; more

Table 2: TNet architecture used on ImageNet (Section 5.1 in our paper). BagNet-77 baseline corresponds to the feature extraction module followed by the classification module. For simplicity, we provide only the spatial dimensions (without the channel dimension) of the feature extraction module's output. The location module receives two inputs and combines them into a single input feature map of size $5 \times 5 \times 1538$ (see Section 2.1.4). The positional encoding module receives a feature vector and a positional encoding vector, and concatenates them to an $1 \times 1024$ input vector (see Section 2.1.5).

| Module | Layer/Block Type | Kernel Size | #Output Channels | Stride | Padding | Activation | Output Size | Receptive Field |
|---|---|---|---|---|---|---|---|---|
| | Input | - | - | - | - | - | $77 \times 77$ | - |
| | Conv | $3 \times 3$ | 64 | 1 | VALID | Leaky ReLU | $75 \times 75$ | $3 \times 3$ |
| | ConvBlock | $3 \times 3$ | 256 | 2 | SAME | - | $38 \times 38$ | $5 \times 5$ |
| | ConvBlock | $3 \times 3$ | 256 | 1 | SAME | - | $38 \times 38$ | $9 \times 9$ |
| | ConvBlock | $1 \times 1$ | 256 | 1 | SAME | - | $38 \times 38$ | $9 \times 9$ |
| | ConvBlock | $3 \times 3$ | 512 | 2 | SAME | - | $19 \times 19$ | $13 \times 13$ |
| | ConvBlock | $3 \times 3$ | 512 | 1 | SAME | - | $19 \times 19$ | $21 \times 21$ |
| Feature Extraction | ConvBlock ($\times 2$) | $1 \times 1$ | 512 | 1 | SAME | - | $19 \times 19$ | $21 \times 21$ |
| | ConvBlock | $3 \times 3$ | 1024 | 2 | VALID | - | $9 \times 9$ | $29 \times 29$ |
| | ConvBlock | $3 \times 3$ | 1024 | 1 | SAME | - | $9 \times 9$ | $45 \times 45$ |
| | ConvBlock ($\times 4$) | $1 \times 1$ | 1024 | 1 | SAME | - | $9 \times 9$ | $45 \times 45$ |
| | ConvBlock | $3 \times 3$ | 2048 | 1 | VALID | - | $7 \times 7$ | $61 \times 61$ |
| | ConvBlock | $3 \times 3$ | 2048 | 1 | SAME | - | $7 \times 7$ | $77 \times 77$ |
| | ConvBlock | $1 \times 1$ | 2048 | 1 | SAME | - | $7 \times 7$ | $77 \times 77$ |
| | Conv | $1 \times 1$ | 512 | 1 | SAME | Leaky ReLU | $7 \times 7$ | $77 \times 77$ |
| | GAP | - | 512 | - | - | - | $1 \times 1$ | - |
| | Input | - | - | - | - | - | $5 \times 5 \times 1024$, $1 \times 1 \times 512$ | - |
| Location | Conv | $1 \times 1$ | 512 | 1 | SAME | Leaky ReLU | $5 \times 5 \times 512$ | - |
| | Conv | $1 \times 1$ | 1 | 1 | SAME | - | $5 \times 5 \times 1$ | - |
| | $L_2$ Normalization | - | 25 | - | - | - | $1 \times 25$ | - |
| | Softmax | - | 25 | - | - | - | $1 \times 25$ | - |
| Positional Encoding | Input | - | - | - | - | - | $1 \times 512$, $1 \times 512$ | - |
| | Fully Connected | - | 512 | - | - | - | $1 \times 512$ | - |
| Classification | Input | - | - | - | - | - | $1 \times 512$ | - |
| | Fully Connected | - | 1000 | - | - | - | $1 \times 1000$ | - |

details are provided in Section 2.1.4. The classification module is a linear layer, as in Table 3. The number of output nodes is equal to the number of classes; 200 for CUB-200-2011 [24], and 555 for NABirds [22].

In Table 4, we provide the feature weighting module of TNet-B0. The same design is followed for the other TNet-B$i$ models as well, with output channels scaled according to different feature extraction modules. The input to the feature weighting module is of variable size, as it depends on the number of attended locations. More details are provided in Section 2.1.6.

### 2.1.4 Location module

In Tables 2 and 3, we provide two different implementations of the location module. In Table 2, location module receives two inputs. The first one is a feature map of size $5 \times 5 \times 1024$, which originates from an intermediate layer of the feature extraction module. The spatial dimensions of the feature map are equal to the dimensions of the candidate location grid. Each $1 \times 1 \times 1024$ vector of the feature map, describes the image region within the corresponding grid cell.

To achieve this, we aim for the receptive field of each pixel in the feature map to align with the image region that it is supposed to describe. In the specific architecture of Table 2, we assume a $5 \times 5$ grid of overlapping cells, and an input to the feature extraction module of fixed size $77 \times 77$ px. Each grid cell occupies $34.375\%$ of the corresponding input dimension. Based on that, when the $5 \times 5$ grid is superimposed onto the $77 \times 77$ px input, each cell is approximately of size $27 \times 27$ px.

The layer of the feature extraction module with the closest receptive field size, is in the 8-th ConvBlock with $29 \times 29$ px. However, the effective receptive field size is usually smaller that the actual receptive

Table 3: TNet architecture used on fMoW (Section 5.2 in our paper). EfficientNet-B0 baseline corresponds to the feature extraction module followed by the classification module. For simplicity, we provide only the spatial dimensions of the feature extraction module's output. The location module receives the downsampled output feature map of the 8-th MBConv block as input (receptive field of $147 \times 147$ px). The positional encoding module receives a $1 \times 320$ positional encoding vector that is projected to $1 \times 1280$, and then it is added to the second input of the module, which is a $1 \times 1280$ feature vector.

| Module | Layer/Block Type | Kernel Size | #Output Channels | Stride | Batch Norm | Activation | Output Size | Receptive Field |
|---|---|---|---|---|---|---|---|---|
| | Input | - | - | - | - | - | $224 \times 224$ | - |
| | Conv | $3 \times 3$ | 32 | 2 | $\checkmark$ | SiLU | $112 \times 112$ | $3 \times 3$ |
| | MBConv1 | $3 \times 3$ | 16 | 1 | - | - | $112 \times 112$ | $7 \times 7$ |
| | MBConv6 | $3 \times 3$ | 24 | 2 | - | - | $56 \times 56$ | $11 \times 11$ |
| | MBConv6 | $3 \times 3$ | 24 | 1 | - | - | $56 \times 56$ | $19 \times 19$ |
| | MBConv6 | $5 \times 5$ | 40 | 2 | - | - | $28 \times 28$ | $35 \times 35$ |
| | MBConv6 | $5 \times 5$ | 40 | 1 | - | - | $28 \times 28$ | $67 \times 67$ |
| **Feature Extraction** | MBConv6 | $3 \times 3$ | 80 | 2 | - | - | $14 \times 14$ | $83 \times 83$ |
| | MBConv6 | $3 \times 3$ | 80 | 1 | - | - | $14 \times 14$ | $115 \times 147$ |
| | MBConv6 | $3 \times 3$ | 80 | 1 | - | - | $14 \times 14$ | $147 \times 115$ |
| | MBConv6 | $5 \times 5$ | 112 | 1 | - | - | $14 \times 14$ | $211 \times 211$ |
| | MBConv6 ($\times 2$) | $5 \times 5$ | 112 | 1 | - | - | $14 \times 14$ | $339 \times 339$ |
| | MBConv6 | $5 \times 5$ | 192 | 2 | - | - | $7 \times 7$ | $403 \times 403$ |
| | MBConv6 ($\times 3$) | $5 \times 5$ | 112 | 1 | - | - | $7 \times 7$ | $787 \times 787$ |
| | MBConv6 | $3 \times 3$ | 320 | 1 | - | - | $7 \times 7$ | $851 \times 851$ |
| | Conv | $1 \times 1$ | 1280 | 1 | $\checkmark$ | SiLU | $7 \times 7$ | $851 \times 851$ |
| | GAP | - | 1280 | - | - | - | $1 \times 1$ | - |
| | Input | - | - | - | - | - | $3 \times 3 \times 80$ | - |
| | Conv | $1 \times 1$ | 80 | 1 | - | SiLU | $3 \times 3 \times 80$ | - |
| | SE-(80, 0.5) | - | 80 | - | - | - | $3 \times 3 \times 80$ | - |
| **Location** | Conv | $1 \times 1$ | 80 | 1 | - | SiLU | $3 \times 3 \times 80$ | - |
| | Conv | $1 \times 1$ | 1 | 1 | - | - | $3 \times 3 \times 1$ | - |
| | $L_2$ Normalization | - | 9 | - | - | - | $1 \times 9$ | - |
| | Softmax | - | 9 | - | - | - | $1 \times 9$ | - |
| | Input | - | - | - | - | - | $1 \times 320$ | - |
| **Positional Encoding** | Fully Connected | - | 1280 | - | - | - | $1 \times 1280$ | - |
| | Input | - | - | - | - | - | $1 \times 1280$ | - |
| | Add | - | 1280 | - | - | SiLU | $1 \times 1280$ | - |
| **Classification** | Input | - | - | - | - | - | $1 \times 1280$ | - |
| | Fully Connected | - | 62 | - | - | - | $1 \times 62$ | - |

Table 4: The feature weighting module of TNet-B0 (Section 5.3 in our paper). The input consists of the $N$ feature vectors extracted while attending to a sequence of $N - 1$ locations; $N - 1$ vectors are extracted from the attended locations, and a feature vector from the downscaled version of the whole image (1st processing level). The module first calculates $N$ weights that sum up to 1, and then, it uses them to perform a weighted average of the $N$ input feature vectors.

| Module | Layer/Block Type | Kernel Size | #Output Channels | Stride | Padding | Activation | Output Size |
|---|---|---|---|---|---|---|---|
| | Input | - | - | - | - | - | $1 \times N \times 1280$ |
| **Feature Weighting** | SE-(1280, 0.25) | - | 1280 | - | - | - | $1 \times N \times 1280$ |
| | Conv | $1 \times 1$ | 1 | 1 | SAME | - | $1 \times N \times 1$ |
| | Softmax | - | $N$ | - | - | - | $1 \times N$ |
| | Multiply | - | 1280 | - | - | - | $1 \times 1280$ |

field size [14], as a result, we pick the output feature map of the 13-th ConvBlock with receptive field $45 \times 45$ px. The spatial dimensions of this feature map are $9 \times 9$, and we need to downsample it to $5 \times 5$ px. To this end, we calculate the image level coordinates of the receptive field centers of

the feature map pixels, and we pick the 25 of them with receptive fields that better align with the assumed candidate image regions. Based on our previous remarks about the effective receptive field size, we don't consider perfect alignment to be crucial.

The second input to the location module provides contextual information, and it is the output feature vector of the feature extraction module. This vector is of size $1 \times 1 \times 512$, and we concatenate it across the channel dimension at each spatial position of the input feature map, increasing its size to $5 \times 5 \times 1536$.

We pass the combined feature map through two $1 \times 1$ convolutional layers. The first one fuses the features with the concatenated context. The second one projects each fused vector to a logit value, which represents the relative importance of the corresponding candidate location.

We use the same weights to estimate the importance of each candidate location ($1 \times 1$ convolutions). We don't want to use different sets of weights (e.g., to have 25 output heads [21]), because this doesn't allow information learned in one location to transfer to other locations. Also, less attended locations (e.g., corners) could lead to a partially trained model with erratic behavior.

The downside of $1 \times 1$ convolutions is that they disregard spatial information. To mitigate this problem, we enrich the input tensor (the one of size $5 \times 5 \times 1536$) with positional information according to [27]. In particular, for each spatial location, we calculate horizontal and vertical coordinates in the normalized range $[-1, 1]$. Then, we use 2 linear layers (one for each spatial dimension), to map coordinates into a learned range. The resulting 2-dimensional vectors are concatenated across the channel dimension, resulting to an input feature map of size $5 \times 5 \times 1538$. This is the feature map that we actually pass through the $1 \times 1$ convolutional layers.

The estimated logits are reshaped to a $1 \times 25$ vector, which is first normalized to have $L_2$ norm equal to 1, and then it is passed through a Softmax layer to get the final parameters of the Categorical distribution. The $L_2$ normalization aims to reduce the variance between logits, because we empirically observe that logit values may be too negative, or very close to zero, leading Softmax outputs to be exactly 0, and thus hindering the backpropagation of gradients.

The architecture of the location module in Table 3 is conceptually the same, but has some technical differences. In particular, we provide only one input, the output feature map of the 8-th MBConv block (selected and downsampled according to the process described before). This means that we don't provide the output vector of the feature extraction module as an additional input. The reason is that its size of $1 \times 1280$ results in a parameter-heavy location module, which is antithetical to the very light design of the feature extraction module.

To inject contextual information to the input feature map, we pass it through a squeeze-and-excitation (SE) block [10]. Other than that, we follow the design principles described before. We use $1 \times 1$ convolutions, we augment the SE output feature map with 2-dimensional spatial coordinates' vectors, and we $L_2$ normalize the logits.

In TNet-B$i$ models, location module is implemented as in Table 3 (see Section 2.1.3). However, the input feature map may originate from different layers of the feature extraction module, to account for receptive field differences between backbone networks. For TNet-B0, the location module receives as input the output of the 6-th MBConv block, while for TNet-B1, TNet-B2 and TNet-B3, the input comes from the 7-th MBConv block. Finally, for TNet-B4, the location module receives as input the output of the 8-th MBConv block

### 2.1.5 Positional encoding module

In Tables 2 and 3, we provide two different implementations of the positional encoding module. In both cases, the positional encoding module receives two inputs. The first one is the output feature vector of the feature extraction module. The second input is a vector that encodes positional information about the image region described by the first input. The encoded positional information is 3-dimensional; the first 2 dimensions correspond to spatial coordinates, and the 3rd one to scale.

Given a processing level $l$, we assume that a grid is superimposed onto the input image, where its cells correspond to all possible candidate locations of the level. In the example of Fig. 2 in our paper, in the 1st processing level, the assumed grid consists of a single cell. In the second level ($l = 2$) the gird is $2 \times 2$, and for $l = 3$ the gird is $4 \times 4$.

The spatial coordinates of the grid cells start with $(0, 0)$ in the top left corner, and increase linearly with step 1 both horizontally and vertically. The scale coordinate is equal to $l - 1$. Based on this, each candidate image region has a unique positional triplet $(x, y, s)$, where $x$, $y$ are the spatial coordinates and $s$ is the scale.

We use sine and cosine functions of different frequencies to encode positional triplets $(x, y, s)$ according to [23]. In particular, for positional encodings of size $1 \times N$, we get:

$$P_s(p, \vec{t}) = \sin(p \cdot \left(\frac{1}{100}\right)^{\frac{\vec{t}}{\lfloor N/6 \rfloor}}),$$

$$P_c(p, \vec{t}) = \cos(p \cdot \left(\frac{1}{100}\right)^{\frac{\vec{t}}{\lfloor N/6 \rfloor}}),$$

$$p \in [x, y, s]$$

$$\vec{t} = [0, 1, 2, ... \lfloor N/6 \rfloor]$$

The final positional encoding for triplet $(x, y, s)$, results by concatenating $P_s(x, \vec{t})$, $P_c(x, \vec{t})$, $P_s(y, \vec{t})$, $P_c(y, \vec{t})$, $P_s(s, \vec{t})$ and $P_c(s, \vec{t})$.

The main reason we use these positional encodings (instead of, e.g., learned positional embeddings [5]), is that they can generalize to scales and spatial dimensions of arbitrary size. This is particularly useful for our model, because it has the potential to extend its processing to an arbitrary number of levels.

In Table 2, the positional encoding module concatenates its 2 input vectors, and fuses their information through a linear layer. In Table 3, we implement the positional encoding module differently, because we aim for a relatively smaller number of parameters. To this end, we use positional encodings of 4 times smaller dimensionality compared to the input feature vector (320 instead of 1280). In addition, only the positional encoding is processed by a trainable linear layer. This linear projection brings the positional encoding to the size of the feature vector, while it provides a learned component to the process of encoding position. The projected positional encoding is simply added to the input feature vector, and the outcome is passed through a non-linearity.

### 2.1.6 Feature weighting module

Given a sequence of $N - 1$ attended locations, TNet extracts a total number of $N$ feature vectors; $N - 1$ vectors from the attended locations, and a feature vector from the downscaled version of the whole image (1st processing level). Feature weighting module estimates $N$ weights that sum up to 1, in order to perform a weighted average of the $N$ feature vectors.

As we can see in Table 4, feature weighting module receives $N$ feature vectors as input. Since the number of attended locations varies, the input to the feature weighting module is of variable size as well. In order to calculate the $N$ weights, we first inject contextual information to the $N$ feature vectors through a squeeze-and-excitation block. Then, each of the resulting $N$ vectors, is projected to a logit value through a $1 \times 1$ convolutional layer. The $N$ logits are passed through a Softmax layer to get the final weights. The weighted average is implemented via multiplication with the input feature vectors.

## 2.2 Training

### 2.2.1 Training on ImageNet

To train TNet, we use a single sample ($M = 1$) for the Monte Carlo estimators, and we set $\lambda_f = 0.1$ (Eq. 6). We experimented with different $M$ values, e.g., of 2 and 4, but we observed no significant differences in performance. Since the value of $M$ has a multiplicative effect on the batch size, which leads to considerable increase in training time, we set $M = 1$ in all our experiments. The BagNet-77 baseline is trained by minimizing the cross-entropy classification loss.

For both models we use batches of 64 images, distributed in 4 GPUs. We use the Adam optimizer with the default values of $\beta_1 = 0.9$, $\beta_2 = 0.999$ and $\epsilon = 10^{-8}$. We use xavier initialization [6] for the weights, and zero initialization for the biases. For regularization purposes, we use data augmentation that is very similar to the one used in [16]. In particular, given a training image, we get a random crop

that covers at least $85\%$ of the image area, while it has an aspect ratio between 0.5 and 2.0. Since we provide inputs of fixed size to our networks ($224 \times 224$ px), we resize the image crops accordingly. Resizing is performed by randomly selecting between 8 different methods, which include bilinear, nearest neighbor, bicubic, and area interpolation. Also, we randomly flip the resized image crops horizontally, and we apply photometric distortions [9]. The final image values are scaled in range $[-1, 1]$. Finally, the dropout mentioned in Section 5.1 of our paper, is spatial [20].

Since per-feature regularization plays a crucial role in the performance of TNet, we experimented with a variety of different values for $\lambda_r$ and $\lambda_c$, including 0.1, 0.3, 0.5, 0.7 and 0.9, while $\lambda_r$ and $\lambda_c$ were not always set to be equal. We conducted similar tuning for $\lambda_f$, observing that differences in its value didn't have the impact that those of $\lambda_r$ and $\lambda_c$ had. In the following Sections we report only our final choices for the values of $\lambda_r$, $\lambda_c$ and $\lambda_f$, which led to the best performance.

### 2.2.2 Training on fMoW

We first train TNet with inputs of size $448 \times 448$ px, allowing 2 processing levels. We train for 40 epochs with batches of 64 images (on 4 GPUs), with initial learning rate of 0.001 that drops once by a factor of 0.1. We use the Adam optimizer with its default parameter values, and we follow the weight initialization of [17].

We attend to a fixed number of 2 locations. We use $\lambda_f = 0.1$ and per-feature regularization with $\lambda_c = \lambda_r = 0.2$. We use a single sample for the Monte Carlo estimators.

We use dropout before the linear layer of the classification module with 0.5 drop probability. We use stochastic depth [11] with drop probability that increases linearly to a maximum value of 0.3. We use the data augmentation technique described in Section 2.2.1.

We fine-tune TNet for 10 epochs on images of $896 \times 896$ px, with a fixed number of 2 attended location in the 2nd processing level, and 1 in the 3rd (4 in total). Compared to the previous step, we increase the maximum drop probability of stochastic depth to 0.5, and we set $\lambda_c = \lambda_r = 0.05$. Also, we only use features extracted until the 2nd processing level in per-feature regularization (features extracted in the 3rd processing level are excluded).

We use different input images to train 4 EfficientNet-B0 baselines. For the first baseline we use images cropped according to the bounding box annotations, and resized to $224 \times 224$ px. We train for 65 epochs with batches of 64 images, on 4 GPUs. Our initial learning rate is 0.001, and it drops once by a factor of 0.1. We use the Adam optimizer with its default parameter values, and we follow the weight initialization of [17].

We use dropout before the final classification layer with 0.75 drop probability, and $L_2$ regularization with weight of $10^{-5}$. We use stochastic depth with drop probability that increases linearly to a maximum value of 0.5. We use the data augmentation technique described in Section 2.2.1.

The second baseline is trained on the original images, resized to $224 \times 224$ px. The only difference with the training of the previous baseline is that we train for 60 epochs.

The third baseline is trained on the original images resized to $448 \times 448$ px. We train for 30 epochs with batches of 32 images. We reduce stochastic depth maximum drop probability to 0.3. All other training hyperparameters remain the same.

The fourth baseline is trained on the original images resized to $896 \times 896$ px. We train for 30 epochs with batches of 32 images. We set dropout drop probability to 0.3, and stochastic depth maximum drop probability to 0.2. All other training hyperparameters remain the same.

### 2.2.3 Training on CUB-200-2011 and NABirds

CUB-200-2011 [24] and NABirds [22] are fine-grained classification datasets with images of different bird species. Images from different classes may exhibit very high visual similarity, and as a result, successful classification requires learning subtle discriminative features. To alleviate this problem, we consider the contrastive loss term from [7]:

$$L_{con} = \lambda_{con} \cdot \frac{1}{N^2} \sum_i^N \Big[ \sum_{j:y_i=y_j}^N \big( 1 - \text{cos\_sim}(f_i, f_j) \big) + \sum_{j:y_i \neq y_j}^N \max \big( \text{cos\_sim}(f_i.f_j) - \alpha, 0 \big) \Big] \quad (9)$$

where $N$ is the batch size, $x_k$ is the $k$-th image in the batch, $f_k$ is a feature vector representing $x_k$, $y_k$ is the class label of $x_k$, and cos_sim$(\cdot, \cdot)$ is a function that receives two vectors as input and calculates their cosine similarity. $\alpha$ is a hyperparameter that constitutes a similarity threshold. $\lambda_{con}$ is a hyperparameter that specifies the relative importance of $L_{con}$ within the total loss used for training.

The first term in Equation 9 is used to maximize the similarity between feature vectors that represent images from the same class. The second term is used to not allow the similarity between feature vectors that represent images from different class to exceed $\alpha$.

For all models, we use pre-trained weights that are available in [19]. In particular, we use the weights of EfficientNet models trained with NoisyStudent [25] and RandAugment [3] on ImageNet with extra JFT-300M unlabeled data.

**Training on CUB-200-2011.** We train TNet-B0 on images of size $448 \times 448$ px, for 200 epochs, with batches of 64 images, on 4 NVIDIA Quadro RTX 8000 GPUs. The feature extraction module is initialized with pre-trained weights, while for the rest of the modules we follow the random weight initialization of [17]. For the weights of the feature extraction module we use a learning rate of $10^{-4}$, while for the rest of the weights we use a learning rate of $10^{-3}$. Both learning rates drop once by a factor of $0.1$. We use the Adam optimizer with its default parameter values.

We attend to a fixed number of 5 locations, with processing extended to 2 levels. We use the learning rule of Eq. 3 from our paper, with $\lambda_f = 0.1$, and $\lambda_c = \lambda_r = 0.3$. We use a single sample for the Monte Carlo estimators.

We use dropout [15] before the linear layer of the classification module with $0.75$ drop probability. We use stochastic depth [11] with drop probability that increases linearly to a maximum value of $0.5$. We use $L_2$ regularization with weight of $10^{-4}$. We use contrastive loss with $\lambda_{con} = 100$ and $\alpha = 0.4$.

For data augmentation, given a training image, we get a random crop that covers at least $85\%$ of the image area, while it has an aspect ratio between $0.5$ and $2.0$. We resize the image crop to $448 \times 448$ px by randomly selecting between 8 different resizing methods. We randomly flip the resized image crops horizontally. We don't apply photometric distortions because color is a discriminative feature for bird species. We perform random translation and rotation of the image. The final image values are scaled in the range of $[-1, 1]$.

We train TNet-B1, TNet-B2 and TNet-B3 by following the same training procedure we described for TNet-B0. For TNet-B4, the only differences is that we train for fewer epochs, 125 instead of 200.

We train all EN-B$i$, $i \in \{0, 2, ..., 4\}$ baselines under the same training regime. We train on images of size $448 \times 448$ px, for 200 epochs, with batches of 64 images, on 4 NVIDIA Quadro RTX 8000 GPUs. We initialize all layers with pre-trained weights, except the last fully connected layer, which size depends on the number of output classes. This last output layer is randomly initialized according to [17].

For layers initialized with pre-trained weights, we use a learning rate of $5 \cdot 10^{-5}$, while for the output layer we use a learning rate of $10^{-3}$. Both learning rates drop once by a factor of $0.1$. We use the cross entropy loss, and we add a contrastive loss term with $\lambda_{con} = 100$ and $\alpha = 0.4$. We use Adam optimizer with its default parameter values. We use the regularization and data augmentation methods we described for the TNet-B$i$ models, with the same hyper-parameters as well.

**Training on NABirds.** We train all TNet-B$i$, $i \in \{0, 2, ..., 4\}$ models according to the procedure we followed on CUB-200-2011, and we only change some hyperparameter values. In particular, we train for 100 epochs, and we set $\lambda_{con} = 50$ for the contrastive loss term. We train TNet-B0, TNet-B1 and TNet-B2 with a fixed number of 5 attended locations, while we train TNet-B3 and TNet-B1 with 3 attended locations.

We train all EN-B$i$, $i \in \{0, 2, ..., 4\}$ models according to the procedure we followed on CUB-200-2011. The only difference is that we train for 100 epochs, and we set $\lambda_{con} = 50$ for the contrastive loss term.

Table 5: Detailed results on CUB-200-2011 dataset [24] (Section 5.3 in our paper).

| Model | # Locs | Top-1 Acc. | Top-5 Acc. | FLOPs (B) | Params (M) |
|---|---|---|---|---|---|
| **EfficientNet-B0** | - | 86.49% | 96.82% | 1.55 | 4.31 |
| **EfficientNet-B1** | - | 88.25% | 97.55% | 2.29 | 6.83 |
| **EfficientNet-B2** | - | 88.13% | 97.34% | 2.65 | 8.05 |
| **EfficientNet-B3** | - | 88.42% | 97.38% | 3.88 | 11.09 |
| **EfficientNet-B4** | - | 89.08% | 97.26% | 6.09 | 18.03 |
| **ResNet-50** | - | 84.5% | - | 16.35 | 23.99 |
| **API-Net** [28] | - | 90.0% | - | - | 29 |
| **TransFG** [7] | - | 91.7% | - | - | 86 |
| **TNet-B0** | 5 | 87.75% | 97.27% | 2.32 | |
| | 4 | 87.59% | 97.26% | 1.94 | |
| | 3 | 87.66% | 97.26% | 1.55 | 5.56 |
| | 2 | 87.07% | 97.17% | 1.16 | |
| | 1 | 85.66% | 96.5% | 0.78 | |
| | 0 | 77.84% | 93.5% | 0.39 | |
| **TNet-B1** | 5 | 88.33% | 97.67% | 3.44 | |
| | 4 | 88.18% | 97.55% | 2.86 | |
| | 3 | 88.35% | 97.46% | 2.29 | 8.07 |
| | 2 | 87.56% | 97.20% | 1.72 | |
| | 1 | 85.93% | 96.63% | 1.15 | |
| | 0 | 79.72% | 94.56% | 0.57 | |
| **TNet-B2** | 5 | 88.35% | 97.60% | 3.99 | |
| | 4 | 88.20% | 97.48% | 3.32 | |
| | 3 | 87.80% | 97.27% | 2.66 | 9.55 |
| | 2 | 87.64% | 97.20% | 1.99 | |
| | 1 | 86.16% | 96.81% | 1.33 | |
| | 0 | 80.10% | 94.46% | 0.67 | |
| **TNet-B3** | 5 | 89.35% | 97.88% | 5.84 | |
| | 4 | 89.02% | 97.74% | 4.86 | |
| | 3 | 89.1% | 97.67% | 3.89 | 12.87 |
| | 2 | 88.44% | 97.50% | 2.92 | |
| | 1 | 87.18% | 97.24% | 1.95 | |
| | 0 | 81.64% | 95.41% | 0.97 | |
| **TNet-B4** | 5 | 90.06% | 98.29% | 9.15 | |
| | 4 | 89.97% | 98.21% | 7.63 | |
| | 3 | 89.92% | 98.1% | 6.1 | 20.46 |
| | 2 | 89.27% | 98.02% | 4.58 | |
| | 1 | 87.95% | 97.39% | 3.05 | |
| | 0 | 82.59% | 95.88% | 1.53 | |

## 2.3 Metrics

We calculate the FLOPs of a convolutional layer in the following way:

$$N_{FLOPs} = (C_{in} \cdot k^2) \cdot (H_{out} \cdot W_{out} \cdot C_{out}) \tag{10}$$

where $C_{in}$ is the number of channels in the input feature map, $k \times k$ are the spatial dimensions of the convolutional kernel, $H_{out} \times W_{out}$ is the spatial resolution of the output, and $C_{out}$ is the number of output channels. Each time the kernel is applied, we make $C_{in} \cdot k^2$ multiplications, and we apply the kernel $H_{out} \cdot W_{out} \cdot C_{out}$ times (number of output pixels). For fully connected layers, simply holds $k = 1$ and $H_{out} = W_{out} = 1$.

Equation 10 accounts only for multiplications. If we consider additions as well, the number of FLOPs approximately doubles. We use Eq. 10 because it allows us to calculate FLOPs for our EfficientNet baselines that are in accordance with the FLOPs reported in [17].

We time our models during inference by using 45 sets of 10 batches with 64 images in each batch. For each model, we calculate the average value and the standard deviation among the 45 sets of batches.

Table 6: Detailed results on NABirds dataset [22] (Section 5.3 in our paper).

| Model | # Locs | Top-1 Acc. | Top-5 Acc. | FLOPs (B) | Params (M) |
|---|---|---|---|---|---|
| **EfficientNet-B0** | - | 84.97% | 96.77% | 1.55 | 4.76 |
| **EfficientNet-B1** | - | 86.55% | 97.44% | 2.29 | 7.29 |
| **EfficientNet-B2** | - | 86.79% | 97.50% | 2.65 | 8.55 |
| **EfficientNet-B3** | - | 87.63% | 97.50% | 3.88 | 11.63 |
| **EfficientNet-B4** | - | 87.87% | 97.59% | 6.09 | 18.67 |
| **API-Net** [28] | - | 88.1% | - | - | 29 |
| **TransFG** [7] | - | 90.8% | - | - | 86 |
| **TNet-B0** | 5 | 86.56% | 97.78% | 2.33 | |
| | 4 | 86.49% | 97.72% | 1.94 | |
| | 3 | 86.16% | 97.57% | 1.55 | 6.01 |
| | 2 | 85.56% | 97.22% | 1.16 | |
| | 1 | 83.87% | 96.45% | 0.78 | |
| | 0 | 73.82% | 91.95% | 0.39 | |
| **TNet-B1** | 5 | 87.85% | 98.15% | 3.44 | |
| | 4 | 87.63% | 98.06% | 2.86 | |
| | 3 | 87.20% | 97.89% | 2.29 | 8.52 |
| | 2 | 86.39% | 97.57% | 1.72 | |
| | 1 | 84.65% | 96.77% | 1.15 | |
| | 0 | 76.99% | 93.61% | 0.57 | |
| **TNet-B2** | 5 | 87.52% | 97.92% | 3.99 | |
| | 4 | 87.22% | 97.81% | 3.32 | |
| | 3 | 86.73% | 97.56% | 2.66 | 10.05 |
| | 2 | 85.84% | 97.15% | 1.99 | |
| | 1 | 83.93% | 96.35% | 1.33 | |
| | 0 | 76.58% | 93.15% | 0.67 | |
| **TNet-B3** | 5 | 88.33% | 98.06% | 5.84 | |
| | 4 | 88.26% | 98.00% | 4.87 | |
| | 3 | 87.98% | 97.78% | 3.89 | 13.42 |
| | 2 | 87.47% | 97.56% | 2.92 | |
| | 1 | 86.01% | 96.98% | 1.95 | |
| | 0 | 78.78% | 93.74% | 0.97 | |
| **TNet-B4** | 5 | 88.41% | 98.04% | 9.15 | |
| | 4 | 88.25% | 97.95% | 7.63 | |
| | 3 | 88.07% | 97.76% | 6.1 | 21.09 |
| | 2 | 87.53% | 97.48% | 4.58 | |
| | 1 | 86.12% | 96.82% | 3.05 | |
| | 0 | 79.57% | 93.61% | 1.53 | |

These are the time measurements reported in Tables 1 and 2 of our paper. We measure memory requirements in batches of 64 images, by using the TensorFlow memory profiler. During profiling, we disregard the first processing iterations, to avoid any computational and memory overhead that stems from the creation of the TensorFlow graph. Finally, TensorFlow automatically calculates the number of our models' parameters.

## 2.4 Results

### 2.4.1 Results on fMoW

In Figure 1 we plot the main results from Table 2 in our paper.

### 2.4.2 Results on CUB-200-2011 and NABirds

We present our results on CUB-200-2011 [24] and NABirds [22] datasets in Tables 5 and 6 respectively. API-Net [28] is using a DenseNet-161 backbone [12], and TransFG [7] is using a ViT-B/16 [5].

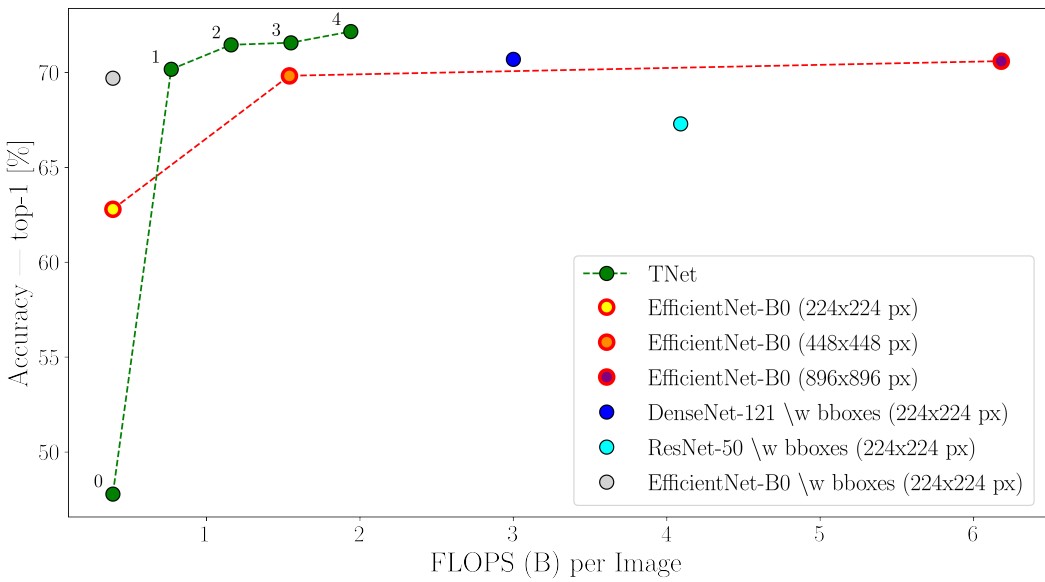

Figure 1: Graphical representation of the main results on fMoW (Table 2 in our paper). Numeric annotations correspond to the number of attended locations.

Table 7: Precision, recall, and image coverage, calculated on ImageNet and fMoW. We use bounding boxes that are available for $544, 546$ training images from ImageNet, and bounding boxes that are available for all testing images of fMoW. Attended locations originate from the 2nd processing level of TNet. Precision on ImageNet is high (small part of the background area is attended), and recall is low (small part of the object of interest is attended); the opposite is observed on fMoW. We attribute this behavior to the bigger size of objects of interest in ImageNet compared to fMoW, and to the smaller attention grid cells that we use on ImageNet. Coverage does not increase linearly with locations' number, because of their overlap. Attending to $\leq 25\%$ of image area suffices to outperform almost all baselines in Tables 1 and 2 of our paper.

| Dataset | # Locs | Precision | Recall | Coverage | Top-1 Acc. |
|---------|--------|-----------|--------|----------|------------|
| **ImageNet** | 1 | 75.52% | 25.62% | 11.71% | 73.12% |
| | 2 | 72.9% | 38.72% | 19.65% | 74.12% |
| | 3 | 70.22% | 47.49% | 26.35% | 74.41% |
| | 4 | 67.66% | 54.84% | 32.46% | 74.58% |
| | 5 | 65.3% | 60.7% | 38.21% | 74.62% |
| **fMoW** | 1 | 31.92% | 86.51% | 25.0% | 70.17% |
| | 2 | 24.37% | 90.06% | 37.91% | 71.46% |

# 3 Attention policy and interpretability

## 3.1 Quantitative analysis

We quantify the localization capabilities of TNet, by using bounding box annotations that are available for ImageNet and fMoW. In particular, given an image and a bounding box, we use the attended image regions at the 2nd processing level, in order to compute precision and recall in the following

way:

$$precision = \frac{|S_{att} \cap S_{bbox}|}{|S_{att}|}, \ S_{att} \neq \emptyset \tag{11}$$

$$recall = \frac{|S_{att} \cap S_{bbox}|}{|S_{bbox}|}, \ S_{bbox} \neq \emptyset \tag{12}$$

where $S_{att}$ is the set of pixels that belong to image regions attended by the location module, $S_{bbox}$ is the set of pixels that belong to the bounding box of the object of interest, and $|S|$ denotes the cardinality of a set $S$. We assume $S_{att} \neq \emptyset$, and $S_{bbox} \neq \emptyset$, meaning that for every image, we attend to at least 1 location, and we have a bounding box with area greater than 0. Both precision and recall take values within $[0, 1]$. Precision measures the percentage of the attended regions' area that overlaps with the bounding box. When precision gets smaller, more background (image area outside the bounding box) is attended. Recall measures the percentage of the bounding box area that is attended. When recall gets smaller, a smaller part from the object of interest (image area inside the bounding box) is attended.

In Table 7, we calculate precision and recall by using bounding boxes that are available for $544, 546$ training images on ImageNet, and bounding boxes that are available for every image in the test set of fMoW. Coverage corresponds to the percentage of the image area that is covered by attended locations. Attended locations usually overlap, and as a result, coverage does not increase linearly with the number of locations. The fact that we use images from the training set of ImageNet, could potentially lead to biased results in Table 7. In an attempt to test this, we calculate coverage on the validation set of ImageNet, since it doesn't require bounding box annotations. We find that coverage values are almost identical to the ones reported in Table 7.

We observe that precision on ImageNet is high (small part of the background area is attended), while recall is low (small part of the object of interest is attended). This means that attended locations mainly occupy a limited area within the bounding boxes of the objects of interest. The opposite is observed in fMoW, with low precision, and high recall. We primarily attribute this difference in behavior to two factors. First, bounding boxes from ImageNet are bigger on average compared to those from fMoW. In particular, on average, a bounding box from ImageNet covers $46\%$ of the image area, while the same metric is $14\%$ for bounding boxes from fMoW. Second, the attention grid cells are smaller on ImageNet (1 location has coverage $11.71\%$ on ImageNet, and $25\%$ on fMoW), and as a result, the attention policy can be more precise.

The fact that bounding boxes from fMoW have an image coverage of only $14\%$, can be used to explain the drop in accuracy that is observed in Table 2 of our paper, when TNet extends processing from $4$ to $6$ locations. Since objects of interest are small and successfully located (high recall), attending to more locations is expected to mostly add uninformative background clutter.

### 3.2 Qualitative examples

In Figure 2 we provide examples of the attention policy on the ImageNet validation set with 3 locations. In Figure 3 we provide examples of the attention policy on the fMoW test set with 2 locations at the 2nd processing level. In Figures 4 and 5 we provide attention policy examples on the validation sets of CUB-200-2011 and NABirds respectively. In both cases, 3 location are attended, and the weights estimated by the feature weighting module are provided as well.

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

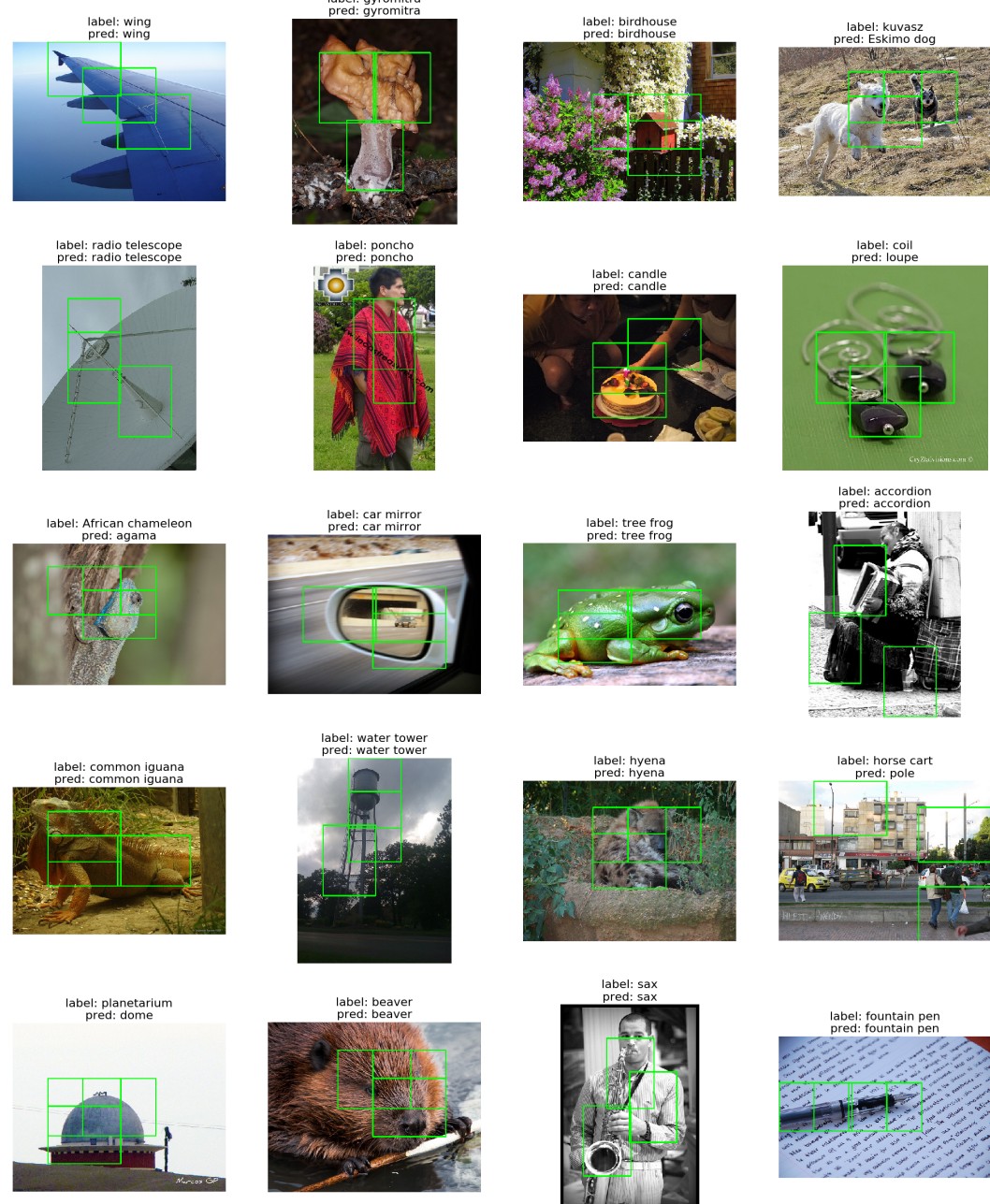

Figure 2: Attention policy examples with 3 locations on the ImageNet validation set. For every image, the correct and predicted labels are provided.

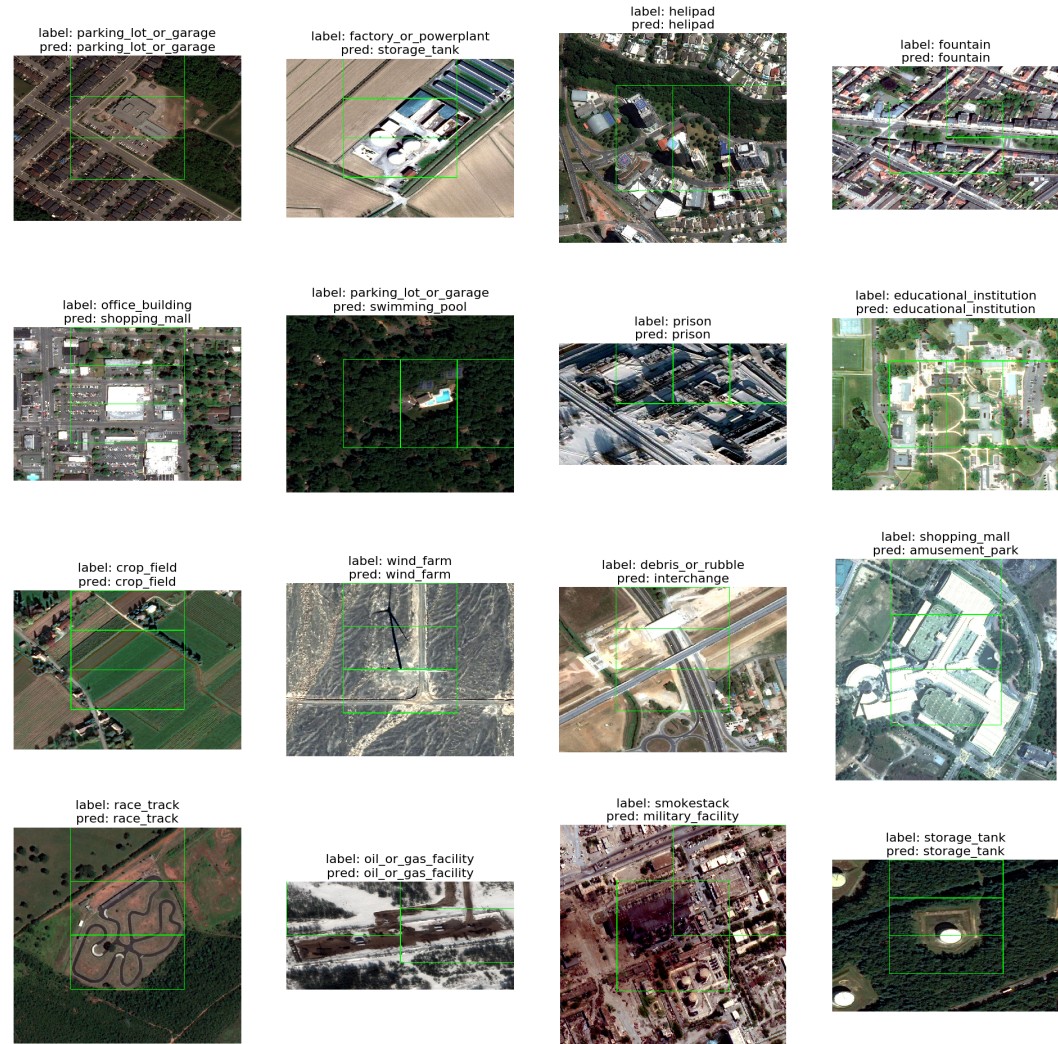

Figure 3: Attention policy examples with 2 locations (2nd processing level) on the fMoW test set. For every image, the correct and predicted labels are provided.

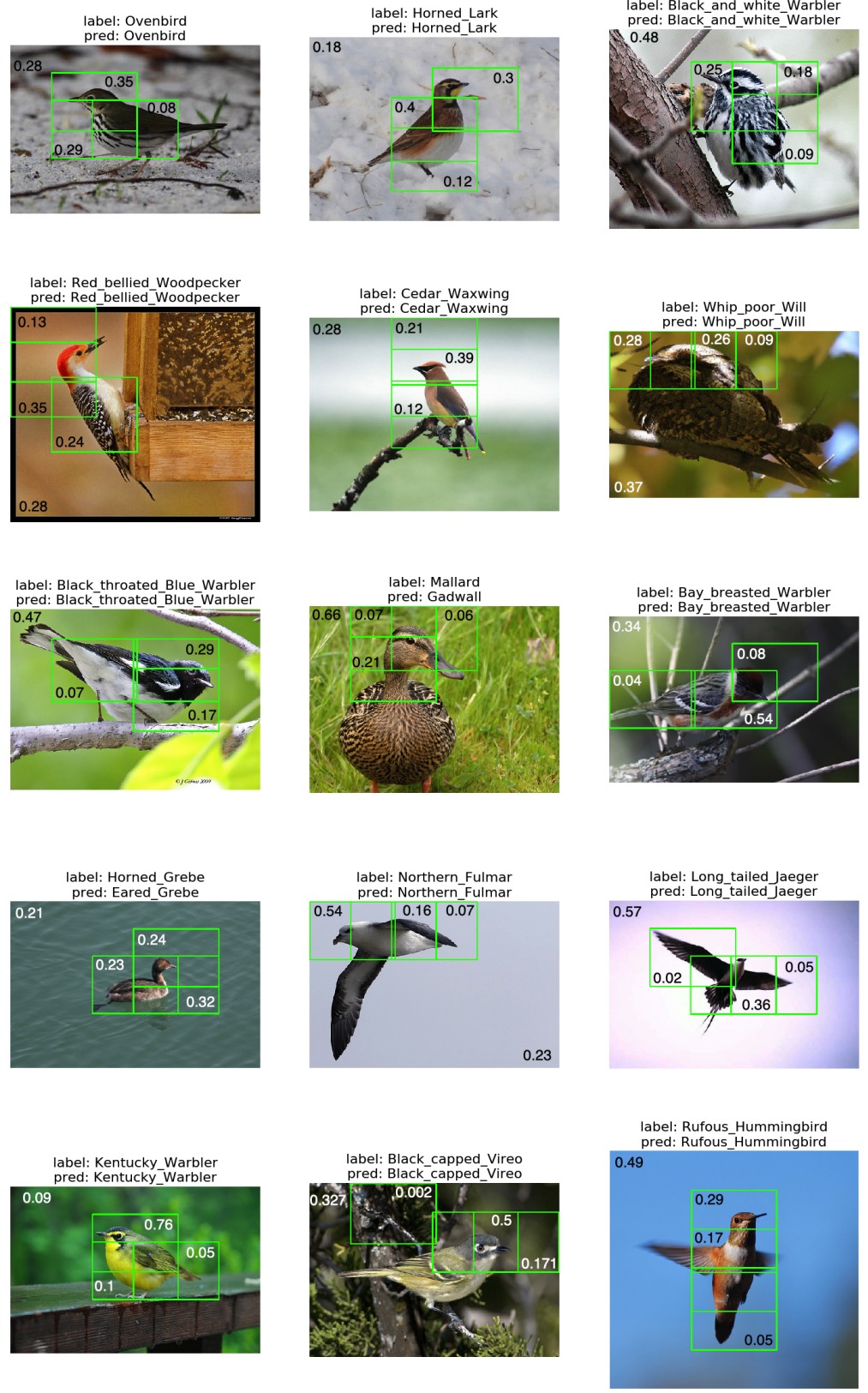

Figure 4: Examples from attention policy learned on CUB-200-2011. Numeric annotations correspond to weights predicted by the feature weighting module, for the top 3 locations and the downscaled version of the whole image (1st processing level). Weights sum up to 1. For every image, the correct and predicted labels are provided.

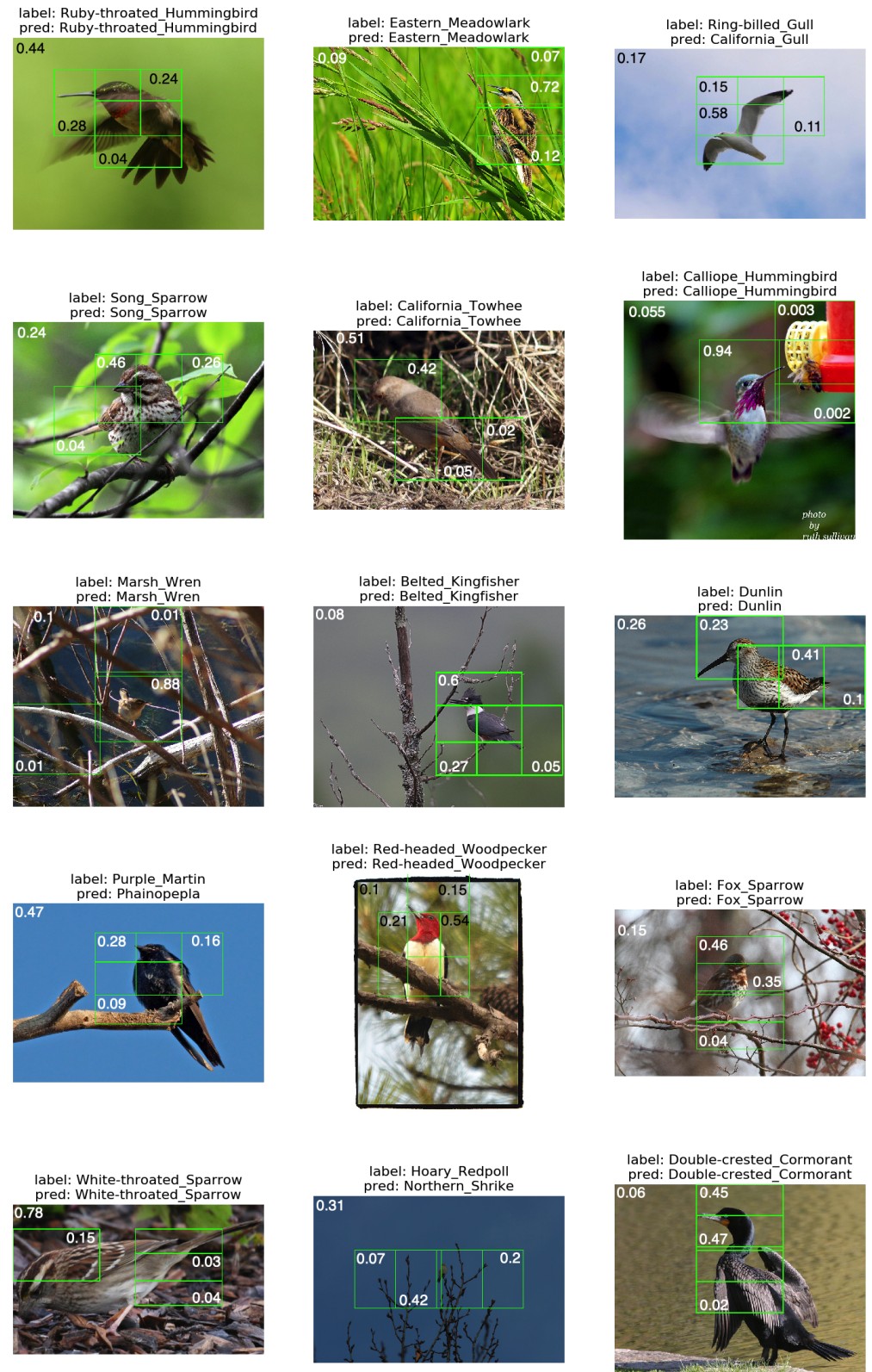

Figure 5: Examples from attention policy learned on NABirds. Numeric annotations correspond to weights predicted by the feature weighting module, for the top 3 locations and the downscaled version of the whole image (1st processing level). Weights sum up to 1. For every image, the correct and predicted labels are provided.