# OpenReview forum: "Hard-Attention for Scalable Image Classification"
_NeurIPS.cc/2021/Conference — NeurIPS 2021 Poster_

### Official Review · Reviewer_YvBo · 2021-07-10

**Rating:** 7
**Confidence:** 3

**Summary:**

High-resolution image classification is a hard problem for DNN due to resource constraints. The authors propose a multi-scale hard-attention architecture, TNet, for high-resolution information, by only visiting the most informative image regions, to achieve better accuracy vs complexity tradeoff. The architecture achieves a better tradeoff than other baselines on both ImageNet and fMoW datasets.

**Limitations And Societal Impact:**

The method proposed could improve the efficiency of the image classification task, and provides the researchers a method to utilize the information in high-resolution images unequally. However, the accuracy of the model under extreme computation limitation is not as good as other baseline models as mentioned above, which constraints the application of the method on limited resources.

**Main Review:**

The authors propose TNet, which leverages the high-resolution information with low computation complexity, using a multi-scale hard-attention architecture.  The paper is clearly written, providing good motivation for the proposed method.

**Strengths**
1. The TNet offers a better trade-off between accuracy and complexity, with hard attention and only using part of the information in the image by attending to the limited number of image regions.
2. The method is orthogonal to the backbone model and can be applied to different models.
3. According to experiment results, the model provides a better trade-off between accuracy and FLOPs, as well as the inference time comparing to the SOTA.
4. The design of selecting important regions with attention probability hierarchically utilizing multi-scale information of the high-resolution image.
5. The reinforcement rule introduced to solve the not differentiable problem by selecting locations is interesting and novel.

**Weaknesses**
1. The experiments in Figure 3 do not indicate improvement of the TNet against the EfficientNet, especially for B2 version. It is a bit contradict with the significant improvement in Table 2. Also, the accuracy of the TNet for very small FLOPs (0.39B) is extremely bad on the fMoW comparing to the baselines in Table 2.
2. In the ablation study, the authors mention that the per-feature regularization may have a similar impact as cropping-based data augmentation. Though they clarify the difference between the two methods, it would be great to see an empirical comparison of random cropping and the per-feature regularization.

Typos
1. 'Our goal it to' in line 26.

**Time Spent Reviewing:**

4 hours

---

> ### Author Response · Authors · 2021-08-06
> **Response to Reviewer YvBo**
>
> Dear Reviewer,
>
> Thank you very much for your comments. You raise a number of important points, please see our responses below:
>
> 1. “The experiments in Figure 3 do not indicate improvement of the TNet against the EfficientNet, especially for B2 version. It is a bit contradict with the significant improvement in Table 2.”
>
> We would like to point to 2 differences between the experiments in Table 2 and Figure 3, in an attempt to explain differences in behavior between TNet and the EfficientNet baselines. First, we note that in Figure 3, processing extends to 2 levels (input resolution of 448x448 px), while in Table 2, processing extends to 3 levels (resolution of 896x896 px). This is important in terms of computational efficiency, because as we process images of higher resolution, we expect TNet to provide more significant computational gains compared to the baselines, as happens in Table 2.
> Second, to explain the smaller differences in accuracy between TNet and the baselines, we hypothesize that the model pre-training is partly responsible. In particular, only the feature extraction module of TNet is initialized with pre-trained weights, while the rest of the modules are randomly initialized. Our hypothesis is that if we had pre-trained the whole TNet architecture on ImageNet, and used these weights to fine-tune on the downstream tasks, the performance of TNet would have been even better.
> Finally, we would like to mention that factors in the observed variance in behavior, may merely originate from differences in the nature of the data. This is one of the primary reasons we evaluated our method on a number of datasets with diverse content, and we plan to keep doing this in follow-up work, e.g., by experimenting on high-resolution medical images.
>
> 2. “The accuracy of the TNet for very small FLOPs (0.39B) is extremely bad on the fMoW comparing to the baselines in Table 2.”
>
> Thank you for pointing this out. Since we focused on processing higher-resolution images, we did not pay too much attention to the suboptimal performance on just the low-resolution context. However, we are confident that there is a simple solution to this problem. In particular, in the context of per-feature regularization, we perform separate predictions by using feature vectors from each attended location (1st term in Equation 2b). We could boost the performance on the 1st processing level, by purposely multiplying the corresponding feature vector with a weighting hyperparameter that would increase its importance during backpropagation. We are willing to include your observation in our limitations, and future work.
>
> 3. “In the ablation study, the authors mention that the per-feature regularization may have a similar impact as cropping-based data augmentation. Though they clarify the difference between the two methods, it would be great to see an empirical comparison of random cropping and the per-feature regularization.”
>
> Thank you very much for your suggestion. We would like to note that in our experiments on ImageNet, we were using random cropping for data augmentation, as we detail in Section 2.2.1 (lines 193-195) of the supplementary material. Based on that, in our ablation study in Section 5.5 (lines 284-287), when we don’t use per-feature regularization, random cropping is still in use, and it doesn’t prevent the significant discrepancy in accuracy, compared to the model trained with per-feature regularization. Though, the model trained with per-feature regularization is using random cropping as well. As a result, our explanation could provide some hints, but a definitive answer would require a separate experiment.

---

> > ### Comment · Reviewer_YvBo · 2021-09-02
> > **Thank you for the response**
> >
> > I appreciate the response from the authors. I enjoyed reading the paper and the idea. The authors' response addressed some of my concerns, including the different results in Table 2 and Figure 3. Most of the responses are based on hypotheses without actual experimental results, which I totally understand due to the time limit, but I will not increase my score.

---

### Official Review · Reviewer_PX35 · 2021-07-14

**Rating:** 7
**Confidence:** 4

**Summary:**

The authors present a hard attention mechanism, called  TNet,  for hi-res image classification.  The methods employ a resolution cascade, where a feature network attends to only a few number of locations of each resolution. The non-differentiable attention sampling is optimized with a Reinfore-like learning rule. All feature vectors are simply pooled for final classification, thus avoiding overhead from recurrent processing.
The authors compare their algorithm with two other hard-attention methods, Saccader and DRAM, and one fully convolutional architecture, BagNet-77 on ILSVRC 2012 ImageNet. Results on the opening figure indicate  that TNet is superior to the other methods on the Pareto frontier of accuracy and GFLOPS per image.

**Limitations And Societal Impact:**

The authors address limitations of their model in Sec. 6. I would add that a further limitation is that, by construction, the model can only attend hi-res regions that have been attended to in low-res too (I'm not sure if that's meant with *"scale-space traversal to be bi-directional"*).

I'm okay with their perspective of the broader impact being in increased resource efficiency, given that in a broader study TNet still can push forward the Pareto frontier of Fig. 1b.

**Main Review:**

## Originality
The idea of doing hard attention is not new, and as such, the TNet architecture does not come with too many big surprises. Please note, that this is not a criticism. What makes the work original and distinctive is how the authors designed the architecture to benefit from multi-resolution hard (and thus sparse) attention, and, at the same time, maintain computational efficiency. That efficiency is achieved imho, by (partially, on every resolution) parallel processing of feature extraction, and by a not overly complicated attention sampling mechanism. Also the feature aggregation with simple averaging is certainly due to  computational efficiency rather than model expressiveness.

## Quality
The approach is technically sound. The derivations of the learning rules follow the line of Reinforce. New and surprising to me was to replace $p(y_i|l^i,x_i,w)$ by an indicator function (line 140-141). Although this sounds plausible, I'd loved to get some motivation for it (or a reference).

What's missing is an exploration of the edge cases of the hyper parameters $\lambda_f$ and $\lambda_r,\lambda_c$ (except for $\lambda_r=\lambda_c=1$ in Sec. 5.5).  In your experiments $\lambda_f=0.1$ is not big, so it could be, in principle, that the second term of the learning rules doesn't actually matter. Same for $\lambda_f,\lambda_r$, could be that $L_F^s$ (2a) is actually irrelevant.
Showing that there is a sweet spot $0<\lambda_{\\{f,r,c\\}}<1$ would remove such concerns.

## Clarity
The paper is well written and easy to follow. I found it extremely fast to get the key message, just from looking at Figs. 1 and 2.
The authors might consider moving some material from the supplementary to the main paper, in order to make the main paper self contained:
* line 140: What is the baseline $b$. Please put supplementary (5) to the main paper.
* Sec. 5.4, Fig. 4: In order to judge the results, it is necessary to know the *feature weighting module*, which is only explained in the appendix.
*  What is the value of $M$? This linearly affects the GFLOPS/image, no? In the appendix (Sec. 1.2) I've found that actually no sampling is done. This should be elaborate in the main paper, as it may affect the correctness of the learning rule.

For all results, please indicate whether the GFLOPS are with or without backward processing. Event better, provide both numbers.

## Significance
I think the work is potentially significant, as hires image processing is a real application. The authors could have added to the significance by addressing a really large scale problem (896x896 is not exactly gigantic). Also, the paper could be stronger by not just comparing with other hard-attention models, but also any other efficient image classification approaches.
If TNet would be still superior, then it would give a clear signal to practitioners to focus on this method.




**Time Spent Reviewing:**

5

---

> ### Author Response · Authors · 2021-08-06
> **Response to Reviewer PX35**
>
> Dear Reviewer,
>
> Thank you very much for your comments. We would like to provide detailed explanations to your comments :
>
> 1. “The approach is technically sound. The derivations of the learning rules follow the line of Reinforce. New and surprising to me was to replace $p(y|l^i, x_i,w)$ by an indicator function (line 140-141). Although this sounds plausible, I'd loved to get some motivation for it (or a reference).”
>
> The use of the indicator function $R$ was adopted from [1], which is the work that we base the derivation of our learning rule as well (we cite this work both in our main manuscript, and in the supplementary material where we detail the derivation of the learning rule, but we didn’t explicitly connect it to the indicator function $R$). The purpose of this choice is to reduce the variance in the second term of our learning rule, because $\log{p(y_i|l^{i}, x_i, w)}$ takes values within the range $[-\infty, 0]$, while the indicator function corresponds to the range $[0, 1]$. As a result, $R$ helps to stabilize training, which tends to be an issue in reward-based learning methods.
>
> [1] Ba, Jimmy, Volodymyr Mnih, and Koray Kavukcuoglu. "Multiple object recognition with visual attention." arXiv preprint arXiv:1412.7755 (2014).
>
> 2. “What's missing is an exploration of the edge cases of the hyper parameters $\lambda_f$ and $\lambda_r$, $\lambda_c$ (except for $\lambda_r=\lambda_c=1$ in Sec. 5.5). In your experiments $\lambda_f=0.1$ is not big, so it could be, in principle, that the second term of the learning rules doesn't actually matter. Same for $\lambda_f$, $\lambda_r$ could be that $L_F^s$ (2a) is actually irrelevant. Showing that there is a sweet spot $0 < \lambda_{\{ f,r,c \}} < 1$ would remove such concerns.”
>
> It is correct that we don’t explicitly mention the different hyperparameter values that we tested, and we only detail our final choices. However, we have experimented with a variety of different values for $\lambda_r$ and $\lambda_c$, and the reported ones are those that lead to the best performance. In particular, we experimented with $\lambda_r$ and $\lambda_c$ that were equal to different values, including $0.1$, $0.3$, $0.5$, $0.7$, and $0.9$, while $\lambda_r$ and $\lambda_c$ were not always equal with each other. Through this experimentation, we concluded that both $L_F^s$ and $L_F^k$ are important for reaching the reported performance. In addition, we experimented with values of $\lambda_f$, concluding to $0.1$, even if differences in its value didn’t have the impact that those of $\lambda_r$ and $\lambda_c$ had. We note that $\lambda_f$ can not be very low, because the second term of Equation 1 (and in Equations 2a and 2b) is responsible for training the attention policy. We are willing to include details about the process of hyperparameter tuning in our supplementary material.
>
> 3. “line 140: What is the baseline $b$. Please put supplementary (5) to the main paper.
> Sec. 5.4, Fig. 4: In order to judge the results, it is necessary to know the feature weighting module, which is only explained in the appendix.”
>
> We are willing to make the suggested adjustments to our manuscript.
>
> 4. “What is the value of $M$? This linearly affects the GFLOPS/image, no? In the appendix (Sec. 1.2) I've found that actually no sampling is done. This should be elaborate in the main paper, as it may affect the correctness of the learning rule.”
>
> Yes, $M$ affects the computational load during training, and this is one of the main reasons that we set its value equal to $1$, since our experimentation with other values, including $2$ and $4$, didn’t lead to a considerable improvement in performance. The actual value of $M$ does not affect the derivation of the learning rule, but we are willing to update our manuscript, in order to clarify that the value of $M$ was always set to $1$; a choice that was motivated by empirical evidence.
>
> 5. “For all results, please indicate whether the GFLOPS are with or without backward processing.”
>
> We follow the convention in the literature, and we measure FLOPs that correspond to inference. We are willing to update our manuscript to clarify this choice.
>
> 6. “The paper could be stronger by not just comparing with other hard-attention models, but also any other efficient image classification approaches. If TNet would be still superior, then it would give a clear signal to practitioners to focus on this method.”
>
> The main reason we didn’t focus on efficient architectures like MobileNets, or on more general strategies that aim to computational efficiency, e.g., model compression, is because such approaches are orthogonal to our proposed architectures. In particular, efficient models can be incorporated to our TNet architecture in place of the feature extraction module, allowing us to increase the efficiency in processing each attended location.
>
> 7. “The authors address limitations of their model in Sec. 6. I would add that a further limitation is that, by construction, the model can only attend hi-res regions that have been attended to in low-res too (I'm not sure if that's meant with "scale-space traversal to be bi-directional").”
>
> That’s exactly right, this is what we mean with bi-directional traversal.

---

> > ### Comment · Reviewer_PX35 · 2021-08-24
> > **Thank you for your response.**
> >
> > As to 1.: Your explanation in the rebuttal makes sense and was helpful. Perhaps you can free up some space in the paper to include it. At least, please cite the reference in this context. It's a \nocite currently as far as I can see.
> >
> > As to 6.:  *"In particular, efficient models can be incorporated to our TNet architecture in place of the feature extraction module, allowing us to increase the efficiency in processing each attended location."* I don't fully buy this argument. It would mean that any ImageNet classification architecture can get better (on the accuracy vs. GFLOPS Pareto frontier) when using the hard attention approach? That's a bold statement and certainly needs empirical foundation. The question that I ask myself as a reader and practitioner is: "I see those blue dot's in Fig. 1b that dominate the other hard attention methods on the Pareto frontier. Fine. But, where do state of the art models end up, regardless of their internal architecture?" If, TNet or the combination with any other efficient feature extraction module cannot beat them, what's the point in employing hard attention?
> >
> >
> > Apart from that I think that my questions were addressed appropriately, and expect that the promised changes can be implemented without problems.

---

> > > ### Author Response · Authors · 2021-08-28
> > > **Response to Reviewer PX35**
> > >
> > > Dear Reviewer,
> > >
> > > Thank you very much for your comments. We would like to provide further clarifications about point 6. One of the main goals for the design of our method was computational efficiency in high resolution images, and to this end, we utilized hard-attention. In this context, ImageNet is not an ideal dataset for experimentation, because it does not consist of very high resolution images. However, there were already strong hard-attention baselines tested on ImageNet (Saccader and DRAM). Based on that, the purpose of our experiments on ImageNet was to compare TNet against these hard-attention baselines in a fair setting, in order to assess the effectiveness of TNet’s hard-attention mechanism. As a result, we didn’t attempt to push the pareto frontier for all types of classification models, but we concentrated on hard-attention baselines. Then, we extended our experiments to fMoW, where we could process high resolution images, and test the scalability of our model. From a practitioner's perspective, we hope that TNet can be useful in settings with high resolution data, like fMoW.
> > >
> > > In addition, we would like to clarify that the purpose of our initial reply wasn’t to make a bold statement about the capabilities of TNet. Our intention was just to highlight that efficient processing offered by architectures like MobilNets or EfficientNets, can benefit TNet as well, since such architectures can be used as the feature extraction module, e.g., as can be seen in our experiments in Section 5.3.

---

> > > > ### Comment · Reviewer_PX35 · 2021-09-10
> > > > **Rebuttal Conclusion**
> > > >
> > > > I like the paper and the idea. The one point that I proposed to make the paper stronger, namely also comparing with non-hard attention models, was not addressed by the authors. So I would not further increase by initial rating (still voting for accept, though)

---

### Official Review · Reviewer_BkyT · 2021-07-16

**Rating:** 8
**Confidence:** 4

**Summary:**

The paper reduces the computational cost of processing large images by leveraging a multi-scale hard attention mechanism. Starting from a low resolution version of an image, it progressively identifies and zooms in the most discriminative areas of the image. By ignoring most of the irrelevant content of the image, it processes images faster than strong traditional approaches, while achieving higher accuracy than these baselines thanks to a novel regularization mechanism.

**Limitations And Societal Impact:**

The authors mentioned the main limitation of the work, which is that the number of glimpses to attend to needs to be tuned manually to achieve a desired level of accuracy, and needs to be retuned for each dataset to achieve optimal performance. They also mention a number of follow up directions to further strengthen the work.

**Main Review:**

The paper makes 3 contributions:
 * It combines a multi scale approach, such as the one successfully used for StreetView house numbers, with a hard attention mechanism. This is a clever approach and a significant contribution.
 * It introduces a new per-feature regularization mechanism that ensures that non discriminative glimpses are identified and penalized in the loss. This is a very ingenuous idea, and like many brilliant ideas, it is both simple and appears obvious in retrospect (yet nobody has thought of it until now as far as I know).
* The model leverages a novel neural network architecture built around 4 modules working together. The whole architecture is flexible, and can be rebuilt as needed around various convolutional models called backbones in the paper (e.g. Bagnet-77,  EfficientNet). This is solid work.

The combined solution is validated by an extensive empirical evaluation. The most significant results demonstrate that:
 * it very significantly outperforms previous hard attention based approaches (namely DRAM and Saccader) both in terms of FLOPs and accuracy
 * it slightly outperforms the backbone model on which it's build (Bagnet-77) in terms of accuracy but significantly outperforms it in terms of FLOPs.
 * it outperforms Resnet50 and Densenet-151, two strong baselines both in terms of FLOPs and accuracy.

The paper also provides numerous examples of images and the corresponding glimpses that were identified by the policy. A visual inspection of these examples provides convincing evidence that the policy is focusing on what we, as human, would consider the most important parts of each image. However, I would have preferred a less anecdotal approach. Perhaps the authors could, for example, evaluate how masking the glimpses identified by the policy impacts the accuracy of the prediction, and compares this to the impact of masking random parts of the image.

Last but not least, the paper attempts to quantify the impact of their per-feature regularization mechanism. Their analysis demonstrates that this technique is a key element to the improved performance compared to previous hard attention work.

The paper is very well written. It is well organized, which makes it a breeze to follow. It was a pleasure to read. Furthermore, the approach can either deliver increased accuracy given a fixed compute budget, or reduce the compute cost given a fixed accuracy target. This tradeoff can be controlled simply by dialing the number location attended to. As a result, and thanks to its impressive results, I believe the paper will get a lot of attention (pun intended)


**Time Spent Reviewing:**

6

---

> ### Author Response · Authors · 2021-08-06
> **Response to Reviewer BkyT**
>
> Dear Reviewer,
>
> Thank you very much for your comments. We are very glad that you enjoyed reading our work. We would like to provide some additional information, based on one of your comments:
>
> 1. “The paper also provides numerous examples of images and the corresponding glimpses that were identified by the policy. However, I would have preferred a less anecdotal approach. Perhaps the authors could, for example, evaluate how masking the glimpses identified by the policy impacts the accuracy of the prediction, and compares this to the impact of masking random parts of the image.”
>
> This is definitely something we could try, and in general we find great value in quantifying the quality of the attention policy in different ways. In that respect, we have used bounding boxes that are available for 544,546 images from the training set of ImageNet, and bounding boxes that are available for all images from the test set of fMoW, to calculate the precision and recall of the attended locations from our learned policy. This is a quantitative analysis that helped us to verify our qualitative assessment of the attention policy, but we had to omit it from our main paper due to space limitations. However, we acknowledge that we could have included it in our supplementary material, and we are willing to correct this by updating our manuscript accordingly.

---

### Official Review · Reviewer_dSSn · 2021-07-16

**Rating:** 7
**Confidence:** 3

**Summary:**

This paper proposes TNet (Traversal Network), a hard attention architecture for classification of high-resolution images using a pyramid of different resolution versions of the input. The proposed method:
- applies a "Feature extraction module" to a 224 x 224 image
- applies a "Location module" to the spatial output of the previous module to select which sub-regions of the input should be further processed
- recursively applies the same steps to each selected subregion, but at a higher-resolution

Each time the "Feature extraction module" is run, it generates a fixed length feature embedding. These embeddings are passed through a "Positional encoding module" and the resulting embeddings are averaged to create a final fixed length representation of the input. The fixed representation can finally be used in a logistic regression classifier.

As this model must dynamically choose which parts of the image are used in the next iteration, it is not end-to-end differentiable, and thus uses a REINFORCE based learning process with M sampled decisions per image to approximate the loss.

The authors test this method against baseline methods on ImageNet, fMoW, CUB-200-2011, and NAbirds. The baseline methods include Saccader, DRAM, BagNet, and EfficientNet.

**Ethical Concerns:**

-

**Limitations And Societal Impact:**

-

**Main Review:**

This paper's contributions are in 1.) a new hard-attention method that can be applied recursively to high-resolution images, and  2.) benchmarking experiments to show the effectiveness of this method against previous work.

The first contribution is novel and interesting, however the experiments for the second contribution are not convincing. For example, the set of experiments on ImageNet do not show the benefits of the proposed approach. The standard procedure for reporting performance on ILSVRC2012 involves resizing images to 224x224 which is not high-resolution, and there exist baseline models with much higher performance and throughput than BagNet-77 (like the EfficientNets). The authors should consider a scatter plot to show model throughput vs. top-1 accuracy to illustrate the pareto frontier of _all_ types of classification models in addition to Table 1. Currently, it is impossible to evaluate how the proposed method compares to other work on this tradeoff.

The argument for efficient performance in the high-resolution setting is more sustained with the fMoW dataset, as satellite imagery (and medical imagery) can be magnitudes larger than "natural" images. However in this case the authors do not compare to SOTA methods. For example, [1] shows top-1 accuracy results >75%.

The results suggesting that Location modules intelligently select subregions of the imagery are encouraging and should be further explored in the remote sensing or medical imaging contexts where the objects of interest _aren't_ in the center crop of the input.


[1] Minetto, Rodrigo, Maurício Pamplona Segundo, and Sudeep Sarkar. "Hydra: An ensemble of convolutional neural networks for geospatial land classification." IEEE Transactions on Geoscience and Remote Sensing 57, no. 9 (2019): 6530-6541.


Minor points
- How many samples, M, are needed to learn / used? Is this an important hyperparameter?
- 140-142: b and R are not used until equation 2 in the next section and should be described there instead.

**Time Spent Reviewing:**

3

---

> ### Author Response · Authors · 2021-08-06
> **Response to Reviewer dSSn**
>
> Dear Reviewer,
>
> Thank you very much for your comments. You raised a number of concerns that we would like to address:
>
> 1. “The set of experiments on ImageNet do not show the benefits of the proposed approach. The standard procedure for reporting performance on ILSVRC2012 involves resizing images to 224x224 which is not high-resolution, and there exist baseline models with much higher performance and throughput than BagNet-77 (like the EfficientNets).
> The authors should consider a scatter plot to show model throughput vs. top-1 accuracy to illustrate the pareto frontier of all types of classification models in addition to Table 1. Currently, it is impossible to evaluate how the proposed method compares to other work on this trade-off.”
>
> We agree that 224x224 px is a typical image resolution, and that BagNet-77 is not an architecture competitive to models from the EfficientNet family. However, the purpose of our experiments on ImageNet was to create a fair setting, where we could compare our method against the hard-attention models of the Saccader and DRAM to assess the effectiveness of our attention mechanism. Based on that, we didn’t attempt to push the pareto frontier for all types of classification models.
> We would like to further clarify that, even if the resolution of 224x224 px is not generally high, in the context of the Saccader and DRAM which process patches in resolution of 77x77 px, 224x224 px is relatively high. We note that we offer complementary experiments on the fMoW dataset, which is of high-resolution compared to the typical image size of 224x224 px.
> Finally, about creating a scatter plot with throughput vs. top-1 accuracy, we provided actual time measurements in msec/im in both Tables 1 and 2, and we are willing to provide additional corresponding plots should that be more visually-friendly to the reader.
>
> 2. “On fMoW, the authors do not compare to SOTA methods. For example, [1] shows top-1 accuracy results >75%.
> [1] Minetto, Rodrigo, Maurício Pamplona Segundo, and Sudeep Sarkar. "Hydra: An ensemble of convolutional neural networks for geospatial land classification." IEEE Transactions on Geoscience and Remote Sensing 57, no. 9 (2019): 6530-6541.”
>
> There was a challenge conducted on the fMoW dataset, organized by IARPA (https://www.iarpa.gov/challenges/fmow.html). In the context of the challenge, many models, including the cited Hydra [1], focused on the final classification performance and created huge ensembles, which we consider impractical for our purposes, and we excluded them from our setting, as we explicitly mention in lines 218-221.
>
> 3. “How many samples, M, are needed to learn / used? Is this an important hyperparameter?”
>
> We experimented with values of $M$ equal to 1, 2, and 4, but we observed no significant differences in performance. Since $M$ has a multiplicative effect on the batch size, which leads to considerable increase in training time, we used $M=1$ in all our experiments (all hyperparameter values are provided in Appendix 2.2).
>
> 4. “140-142: b and R are not used until equation 2 in the next section and should be described there instead.”
>
> You are right, we should have replaced $\log{p(y_i|l^{i}, x_i, w)}$ in the 2nd term of Equation 1 with $R$. We will update our manuscript accordingly.

---

> > ### Comment · Reviewer_dSSn · 2021-08-20
> > **thanks for the response**
> >
> > Thank you for your response!
> >
> > I agree with your reasoning about the experimental setup (a fair comparison isolating _just_ the hard attention component of the models) and didn't consider this. For example, a SOTA EfficientNet (or something) as the feature extraction model should trivially increase the throughput. Based on this I am increasing my score to 7.
> >
> > I do think that a plot like throughput x top-1 acc. would still be very helpful to put the performance of these models in context for the larger community. It seems that as image sizes increase there will be a point at which it becomes practically useful to use hard attention methods.

---

### Official Review · Reviewer_Mp3T · 2021-07-19

**Rating:** 6
**Confidence:** 3

**Summary:**

The paper proposes a Traversal Network that traverse the image scale-space in a top-down fashion, selecting only the most promising areas to compute features on.

The paper propose learning a simple policy for hard-attention to save computation resources in high-resolution image classification.


**Ethical Concerns:**

The paper doesn't address the possible ethical concerns of working with high-resolution images which could include private information or unexpected details.

**Limitations And Societal Impact:**

The authors only briefly hypothesize the the broader impact of their approach, including the possibility of reduce of resource utilization. However, authors didn't mention any potential negative societal impact of their work.

**Main Review:**

# Originality:
Minor originality, the paper builds on previous work about hard-attention and combine well know RL algorithm REINFORCE to learn a simple policy for attention.

# Quality:
Medium quality, there are multiple baseline missing, especially DNN which are designed to be more efficient, ex. EfficientNet, and which are competitive with the proposed solution in terms of FLOPs and accuracy. It's unclear if the proposed method could be applied to more efficient networks.

There is no comparison with random or fixed locations (ex. center crop in multiple resolutions) as reasonable baselines.


# Clarity:
Overall, the paper is clear and well written.


# Significance:
The significance of the work is low, there is not comparison with strong baselines of architectures designed for efficiency or high-resolution images.

Furthermore, the paper doesn't explore other relaxation methods to propagate gradients via sampling, ex. "Categorical Reparameterization with Gumbel-Softmax".

# UPDATE:
Based on the authors detailed comments and other reviews, I've have updated my score to 6.


**Time Spent Reviewing:**

3 hours

---

> ### Author Response · Authors · 2021-08-06
> **Response to Reviewer Mp3T**
>
> Dear Reviewer,
>
> Thank you very much for your comments. Please see our detailed responses below:
>
> 1. “Minor originality: The paper builds on previous work about hard-attention.”
>
> We agree that hard-attention and selective processing have a long history in the community, and we cite multiple relevant methods in our related work. Though, we would like to emphasize that we don’t directly build on an existing method. We designed our architecture from scratch, with critical differences compared to other methods, e.g., we employ a 3-dimensional policy (exploration of 2D space and scale) that has the potential to extend processing to images of any resolution.
>
> 2. “Minor originality: The paper combines well know RL algorithm REINFORCE to learn a simple policy for attention.”
>
> It is correct that the derivation of our learning rule in Equation 1, is based on work that we explicitly cite in Appendix 1.1. However, we extend our learning procedure with a novel regularization term (per-feature regularization), which is crucial for the performance of the model, as we explain in Section 5.5. Furthermore, besides the training procedure, the quality of the attention policy depends on the design of our location module, which we consider non-trivial, and we provide extensive explanations for all our design choices in Appendix 2.1.4.
>
> 3. “there are multiple baseline missing, especially DNN which are designed to be more efficient, ex. EfficientNet, and which are competitive with the proposed solution in terms of FLOPs and accuracy. It's unclear if the proposed method could be applied to more efficient networks.”
>
> We recognize 2 ways to interpret your comment, and we will try to address them both. In case you mean that we should include baselines like EfficientNets to our experiments in Figure 1 (b) and Table 1, the reason we didn’t do it, is that we wanted to make our model directly comparable to the hard-attention models of Saccader and DRAM, and thus, we adjusted our method to their architectural design (BagNet-77 backbone), and training procedure (images of 224x224 px, patches of 77x77 px). Our intention wasn’t to compete on ImageNet accuracy, but to fairly compare with credible hard-attention models.
>
> The second way we interpret your comment, is that you suggest to use EfficientNets as a baseline for efficient scaling, since EfficientNets, among other families of models, establish a principled way for scaling up architectures. In this case, we would like to make 3 remarks about how computationally efficient DNNs like EfficientNets, relate to our proposed architecture.
> (1) The benefits offered by top-performing architectures like EfficientNets, or by very efficient architectures like MobileNets, are orthogonal to our method, since they can be used as the feature extraction module of the proposed TNet. In our experiments in Sections 5.1, 5.2 and 5.3, we used a number of different backbones for TNet, to demonstrate this compatibility.
> (2) If we follow the compound scaling method suggested by EfficientNets to efficiently scale processing to high-resolution images, we have to create a new model for every new resolution. In contrast, we would like to be able to use a single model that has the ability to dynamically adjust its processing.
> (3) Compound scaling can easily become unsustainable. For example, EfficientNet-B7 which operates on images of 600x600 px, requires 37B FLOPs, while our TNet presented in Table 2, spends 1.94B FLOPs on images of 896x896 px. So, besides the potential accuracy benefits that a model like EfficientNet-B7 can offer, our premise is that scaling up models is an unsustainable strategy when we aim to process high-resolution data, hence, we explore an alternative approach based on selective processing. In more general terms, based on the broader impact that we mention in Section 6, we consider that the amount of visual information that we may want to process is practically unbounded, and as a result, we explore a solution adopted by the human brain, which corresponds to overt attention.
>
> 4. “There is no comparison with random or fixed locations (ex. center crop in multiple resolutions) as reasonable baselines.”
>
> In our early experimentation, we trained models with a random policy by using just the first term of Equation 1 (we essentially just minimized cross-entropy, while the weights of the location module maintained their random initialization). We did this as a sanity check for our learning rule. As expected, the performance of TNet with a random policy was poor, and we didn’t save such models.
> However, even if we are not able to report the accuracy of TNet trained with a random policy, in Section 5.5, we report the accuracy of a model trained without per-feature regularization, which can be seen as a model with sub-optimal policy that acts as an upper bound for TNet with random policy. As we report in Section 5.5, the addition of per-feature regularization significantly improves performance.
>
> 5. “The significance of the work is low, there is no comparison with strong baselines of architectures designed for efficiency or high-resolution images.”
>
> Our stated motivation was to design a model that efficiently leverages high-resolution information, compared to convolutional networks that pay a quadratic increase in computation for the same end. Our baseline selection reflects this goal.
>
> 6. “The paper doesn't explore other relaxation methods to propagate gradients via sampling, ex. "Categorical Reparameterization with Gumbel-Softmax".”
>
> Indeed, the learning rule that we used is not the only way to deal with non-differentiable operations like sampling. However, the approach we followed provided results that we deemed satisfying, and we considered that further exploration of alternative training methods was not of primary importance to validate the effectiveness of our proposed architecture.
>
> 7. “The authors only briefly hypothesize the broader impact of their approach, including the possibility of reduce of resource utilization. However, authors didn't mention any potential negative societal impact of their work.
> The paper doesn't address the possible ethical concerns of working with high-resolution images which could include private information or unexpected details.”
>
> We confirm that we didn’t discuss potential negative societal impact of our work, as we explicitly mentioned in our checklist. The reason is that we didn’t identify anything in our methodology, or application, that would be harmful beyond existing image classification models. For example, we don’t propose a generative model which has straightforward potential implications to societal problems like deep fakes. We are willing to include in our manuscript any potential negative impact or ethical concerns that we failed to recognize.

---

> > ### Comment · Reviewer_Mp3T · 2021-08-19
> > **Thanks for detailed response**
> >
> > Thanks for the detailed response and clarifications around novelty and contribution of the paper.
> >
> > The experiments on ImageNet due to smaller resolution 224x224 prevents Hard attention to fully demonstrate its value, probably in tasks with over 1024x1024 resolution Hard attention could be proven much more useful.
> >
> > Based on the comments and other reviews I've updated my score.

---

### Decision · Program_Chairs · 2021-09-27

**Decision:**

Accept (Poster)

**Comment:**

This work proposes reducing the computational cost of running inference on high resolution images by leveraging a multi-scale attention architecture. The method progressively employs higher resolution imagery to zoom in regions of interest that aid in discriminative training, while ignoring any computation on non-informative regions. The resulting model is tested on ImageNet, fMoW and CUB-200-2011 against several baseline architectures, including  Saccader, DRAM, BagNet, and EfficientNet and shows favorable performance in terms of computational cost versus accuracy. The reviewers did voice some concerns about the lack of testing on a high resolution imagery, which remain unresolved. Overall, the reviewers were favorable with this work, and the concerns were notable, and did not rise to a significant enough level. This paper will be accepted to the conference.